# Exact marginal prior distributions of finite Bayesian neural networks

**Jacob A. Zavatone-Veth**
Department of Physics
Harvard University
Cambridge, MA 02138
`jzavatoneveth@g.harvard.edu`

**Cengiz Pehlevan**
John A. Paulson School of Engineering and Applied Sciences
Harvard University
Cambridge, MA 02138
`cpehlevan@seas.harvard.edu`

## Abstract

Bayesian neural networks are theoretically well-understood only in the infinite-width limit, where Gaussian priors over network weights yield Gaussian priors over network outputs. Recent work has suggested that finite Bayesian networks may outperform their infinite counterparts, but their non-Gaussian function space priors have been characterized only though perturbative approaches. Here, we derive exact solutions for the function space priors for individual input examples of a class of finite fully-connected feedforward Bayesian neural networks. For deep linear networks, the prior has a simple expression in terms of the Meijer $G$-function. The prior of a finite ReLU network is a mixture of the priors of linear networks of smaller widths, corresponding to different numbers of active units in each layer. Our results unify previous descriptions of finite network priors in terms of their tail decay and large-width behavior.

## 1 Introduction

Modern Bayesian neural networks (BNNs) ubiquitously employ isotropic Gaussian priors over their weights [1–22]. Despite their simplicity, these weight priors induce richly complex priors over the network's outputs [1, 4–22]. These function space priors are well-understood only in the limit of infinite hidden layer width, in which they become Gaussian [4–8]. However, these infinite networks cannot flexibly adapt to represent the structure of data during inference, an ability that is key to the empirical successes of deep learning, Bayesian or otherwise [2, 3, 9–14, 16–20, 23, 24]. As a result, elucidating how finite-width networks differ from their infinite-width cousins is an important objective for theoretical study.

Progress towards this goal has been made through systematic study of the leading asymptotic corrections to the infinite-width prior [12, 14–17], including approaches emphasizing the physical framework of effective field theory [13, 14]. However, the applicability of these perturbative approaches to narrow networks, particularly those with extremely narrow bottleneck layers [9], remains unclear. In this paper, we present an alternative treatment of a simple class of BNNs, drawing inspiration from the study of exactly solvable models in physics [25–27]. Our primary contributions are as follows:

35th Conference on Neural Information Processing Systems (NeurIPS 2021).

- We derive exact formulas for the priors over the output preactivations of finite fully-connected feedforward linear or ReLU BNNs without bias terms induced by Gaussian priors over their weights (§3). We only consider the prior for a single input example, not the joint prior over the outputs for multiple input examples, as it can capture many finite-width effects [9, 12]. Our result for the prior of a linear network is given in terms of the Meijer $G$-function, which is an extremely general but well-studied special function [28–32]. The prior of a ReLU network is a mixture of the priors of linear networks of narrower widths, corresponding to different numbers of active ReLUs in each layer.

- We leverage our exact formulas to provide a simple characterization of finite-width network priors (§4). The fact that the priors of finite-width networks become heavy-tailed with increasing depth and decreasing width [21, 22], as well as the asymptotic expansions for the priors at large hidden layer widths [12, 15], follow as corollaries of our main results. Moreover, we show that the perturbative finite-width corrections do not capture the heavy-tailed nature of the true prior.

To the best of our knowledge, our results constitute the first exact solutions for the priors over the outputs of finite deep BNNs. As one might expect from knowledge of even the simplest interacting systems in physics [25–27], these solutions display many intricate, non-Gaussian properties, despite the fact that they are obtained for a somewhat simplified setting.

## 2 Preliminaries

In this section, we define our notation and problem setting. We use subscripts to index layer-dependent quantities. We denote the standard $\ell_2$ inner product of two vectors $\mathbf{a}, \mathbf{b} \in \mathbb{R}^n$ by $\mathbf{a} \cdot \mathbf{b}$. Depending on context, we use $\|\cdot\|$ to denote the $\ell_2$ norm on vectors or the Frobenius norm on matrices.

We consider a fully-connected feedforward neural network $\mathbf{f} : \mathbb{R}^{n_0} \to \mathbb{R}^{n_d}$ with $d$ layers and no bias terms, defined recursively in terms of its preactivations $\mathbf{h}_\ell$ as

$$\mathbf{h}_0 = \mathbf{x}, \tag{1}$$
$$\mathbf{h}_\ell = W_\ell \phi_{\ell-1}(\mathbf{h}_{\ell-1}) \qquad (\ell = 1, \dots, d), \tag{2}$$
$$\mathbf{f} = \phi_d(\mathbf{h}_d), \tag{3}$$

where $n_\ell$ is the width of the $\ell$-th layer (i.e., $\mathbf{h}_\ell \in \mathbb{R}^{n_\ell}$) and the activation functions $\phi_\ell$ act elementwise [2, 3]. Without loss of generality, we take the input activation function $\phi_0$ to be the identity. We consider linear and ReLU networks, with $\phi_\ell(x) = x$ or $\phi_\ell(x) = \max\{0, x\}$ for $\ell = 1, \dots, d-1$, respectively. As we focus on the output preactivations $\mathbf{h}_d$, we do not impose any assumptions on the output activation function $\phi_d$.

We take the prior over the weight matrices to be an isotropic Gaussian distribution [1–14, 16–22], with

$$[W_\ell]_{ij} \underset{\text{i.i.d.}}{\sim} \mathcal{N}(0, \sigma_\ell^2) \tag{4}$$

for layer-dependent variances $\sigma_\ell^2$. Depending on how one chooses $\sigma_\ell$—in particular, how it scales with the network width—this setup can account for most commonly-used neural network parameterizations [24]. In particular, one usually takes $\sigma_\ell^2 = \varsigma_\ell^2/n_{\ell-1}$ for some width-independent $\varsigma_\ell^2$ [2–8, 12, 24]. This weight prior induces a conditional Gaussian prior over the preactivations at the $\ell$-th layer [4–8]:

$$\mathbf{h}_\ell \mid \mathbf{h}_{\ell-1} \sim \mathcal{N}(\mathbf{0}, \sigma_\ell^2 \|\phi_{\ell-1}(\mathbf{h}_{\ell-1})\|^2 I_{n_\ell}), \tag{5}$$

where the prior for the first hidden layer is conditioned on the input $\mathbf{x}$, which we henceforth assume to be non-zero. Thus, the joint prior of the preactivations at all layers of the network for a given input $\mathbf{x}$ is of the form

$$p(\mathbf{h}_1, \dots, \mathbf{h}_d \mid \mathbf{x}) = p(\mathbf{h}_d \mid \mathbf{h}_{d-1}) p(\mathbf{h}_{d-1} \mid \mathbf{h}_{d-2}) \cdots p(\mathbf{h}_1 \mid \mathbf{x}). \tag{6}$$

To perform single-sample inference of the network outputs with a likelihood function $p_l(\mathbf{y} \mid \mathbf{h}_d)$ for some target output $\mathbf{y} = \mathbf{y}(\mathbf{x})$, one must compute the posterior

$$p(\mathbf{h}_d \mid \mathbf{x}, \mathbf{y}) = \frac{p_l(\mathbf{y} \mid \mathbf{h}_d, \mathbf{x}) p_d(\mathbf{h}_d \mid \mathbf{x})}{p(\mathbf{y} \mid \mathbf{x})}, \tag{7}$$

where $p(\mathbf{y} \,|\, \mathbf{x}) = \int d\mathbf{h}_d \, p_l(\mathbf{y} \,|\, \mathbf{h}_d, \mathbf{x}) p_d(\mathbf{h}_d \,|\, \mathbf{x})$ [1, 4–13, 18–20]. Before computing the posterior, it is therefore necessary to marginalize out the hidden layer preactivations $\mathbf{h}_1, \ldots, \mathbf{h}_{d-1}$ to obtain the prior density of the output preactivation $p_d(\mathbf{h}_d \,|\, \mathbf{x})$. Moreover, in this framework, all information about the network's inductive bias is encoded in the prior $p_d(\mathbf{h}_d \,|\, \mathbf{x})$, as the likelihood is independent of the network architecture and the prior over the weights. This marginalization has previously been studied perturbatively in limiting cases [4–9, 12, 14]; here we perform it exactly for any width.

To integrate out the hidden layer preactivations, it is convenient to work with the characteristic function $\varphi_d(\mathbf{q}_d \,|\, \mathbf{x})$ corresponding to the density $p_d(\mathbf{h}_d \,|\, \mathbf{x})$. Adopting a convention for the Fourier transform such that

$$p_d(\mathbf{h}_d \,|\, \mathbf{x}) = \int \frac{d\mathbf{q}_d}{(2\pi)^{n_d}} \exp(i\mathbf{q}_d \cdot \mathbf{h}_d)\varphi_d(\mathbf{q}_d \,|\, \mathbf{x}), \tag{8}$$

it follows from (5) that this characteristic function is given as

$$\varphi_d(\mathbf{q}_d \,|\, \mathbf{x}) = \int \prod_{\ell=1}^{d-1} \frac{d\mathbf{q}_\ell \, d\mathbf{h}_\ell}{(2\pi)^{n_\ell}} \exp\left(\sum_{\ell=1}^{d-1} i\mathbf{q}_\ell \cdot \mathbf{h}_\ell - \frac{1}{2}\sum_{\ell=1}^{d} \sigma_\ell^2 \|\mathbf{q}_\ell\|^2 \|\phi_{\ell-1}(\mathbf{h}_{\ell-1})\|^2\right). \tag{9}$$

We immediately observe that the characteristic function is radial, i.e., $\varphi_d(\mathbf{q}_d \,|\, \mathbf{x}) = \varphi_d(\|\mathbf{q}_d\| \,|\, \mathbf{x})$. As the inverse Fourier transform of a radial function is radial [33], this implies that the preactivation prior is radial, i.e., $p_d(\mathbf{h}_d \,|\, \mathbf{x}) = p_d(\|\mathbf{h}_d\| \,|\, \mathbf{x})$. Moreover, as the prior at any given layer is separable over the neurons of that layer, we can see that $p_d(\mathbf{h}_d \,|\, \mathbf{x})$ has the property that the marginal prior distribution of some subset of $k$ of the outputs of a network with $n_d > k$ outputs is identical to the full prior distribution of a network with $k$ outputs. As detailed in Appendix A, these properties enable us to exploit the relationship between the Fourier transforms of radial functions and the Hankel transform, which underlies our calculational approach.

## 3 Exact priors of finite deep networks

Here, we present our main results for the priors of finite deep linear and ReLU networks, deferring their detailed derivations to Appendices A and B of the Supplemental Material.

### 3.1 Two-layer linear networks

As a warm-up, we first consider a linear network with a single hidden layer. In this case, we can easily evaluate the integral (9) to obtain the characteristic function

$$\varphi_2(\mathbf{q}_2 \,|\, \mathbf{x}) = (1 + \kappa_2^2 \|\mathbf{q}_2\|^2)^{-n_1/2}, \tag{10}$$

where we define the quantity $\kappa_2 \equiv \sigma_1 \sigma_2 \|\mathbf{x}\|$ for brevity. We can now directly evaluate the required Hankel transform to obtain the prior density (see Appendix A.1), yielding

$$p_2(\mathbf{h}_2 \,|\, \mathbf{x}) = \frac{1}{(4\pi\kappa_2^2)^{n_2/2}} \frac{2}{\Gamma(n_1/2)} \left(\frac{\|\mathbf{h}_2\|}{2\kappa_2}\right)^{(n_1-n_2)/2} K_{(n_1-n_2)/2}\left(\frac{\|\mathbf{h}_2\|}{\kappa_2}\right), \tag{11}$$

where $\Gamma$ is the Euler gamma function and $K_\nu(z)$ is the modified Bessel function of the second kind of order $\nu$ [28–30].

Interestingly, we recognize this result as the distribution of the sum of $n_1/2$ independent $n_2$-dimensional multivariate Laplace random variables with covariance matrix $2\kappa_2^2 I_{n_2}$ [34]. Moreover, we can see from the characteristic function (10) that we recover the expected Gaussian behavior at infinite width provided that $\kappa_2^2 \propto 1/n_1$ [4–8], as one would expect from the interpretation of this prior as a sum of i.i.d. random vectors. To our knowledge, this simple correspondence has not been previously noted in the literature, though it provides a succinct explanation of the slight heavy-tailedness of this prior distribution noted by Vladimirova et al. [21, 22]. The fact that the function space prior is heavy-tailed at finite width is a particularly important non-Gaussian feature. These results are plotted in Figure 1.

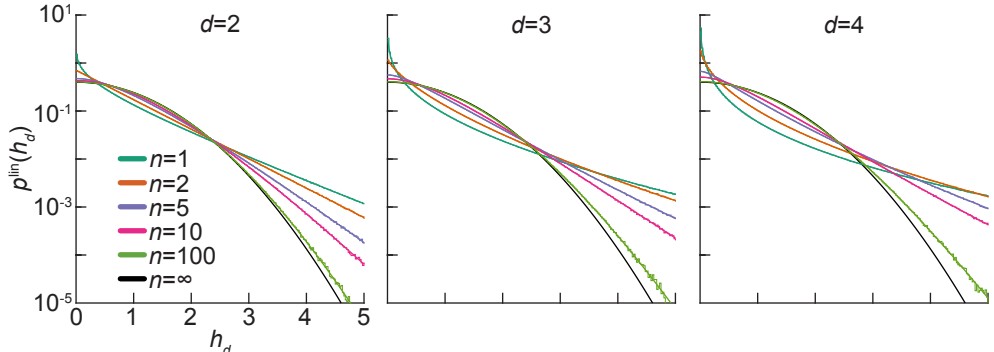

Figure 1: Priors of deep linear networks of depths $d = 2$, $3$, and $4$. In each panel, the prior density is plotted only for positive values of the output preactivation $h_d$, as it is symmetric about zero. For each depth, all hidden layers are of the same width $n$, which is indicated by line color. The black line indicates the Gaussian infinite-width limit discussed in §4.3. Thick lines show the exact priors, while thin jagged lines show experimental estimates from $10^8$ examples. Further details on the numerical methods used to generate these figures are provided in Appendix E.

## 3.2 General deep linear networks

We now consider a general deep linear network. Deferring the details of our derivation to Appendix A, we find that the characteristic function and density of the function space preactivation prior for such a network can be expressed in terms of the Meijer $G$-function [28, 29]. The Meijer $G$-function is an extremely general special function, of which most classical special functions are special cases. Despite its great generality, it is quite well-studied, and provides a powerful tool in the study of integral transforms [28–31]. Its standard definition, introduced by Erdélyi [29], is as follows: Let $0 \leq m \leq q$ and $0 \leq n \leq p$ be integers, and let $a_1, \ldots, a_p$ and $b_1, \ldots, b_q$ be real or complex parameters such that none of $a_k - b_j$ are positive integers when $1 \leq k \leq n$ and $1 \leq j \leq m$. Then, the Meijer $G$-function is defined via the Mellin-Barnes integral

$$G_{p,q}^{m,n}\left(z \left| \begin{array}{c} a_1, \ldots, a_p \\ b_1, \ldots, b_q \end{array}\right.\right) = \frac{1}{2\pi i}\int_C ds\, z^s \frac{\prod_{j=1}^m \Gamma(b_j - s)\prod_{k=1}^n \Gamma(1 - a_j + s)}{\prod_{j=m+1}^q \Gamma(1 - b_j + s)\prod_{k=n+1}^p \Gamma(a_k + s)}, \quad (12)$$

where empty products are interpreted as unity and the integration path $C$ separates the poles of $\Gamma(b_j - s)$ from those of $\Gamma(1 - a_k + s)$ [28, 29]. Expressing the density or characteristic function of a radial distribution in terms of the Meijer $G$-function is useful because one can then immediately read off its Mellin spectrum and absolute moments [28, 29]. Moreover, one can exploit integral identities for the Meijer $G$-function to compute other expectations and transformations of the density [28–31].

With this definition, the characteristic function and density of the prior of a deep linear network are given as

$$\varphi_d^{\text{lin}}(\mathbf{q}_d \,|\, \mathbf{x}) = \gamma_d G_{d-1,1}^{1,d-1}\left(2^{d-2}\kappa_d^2\|\mathbf{q}_d\|^2 \left| \begin{array}{c} 1 - n_1/2, \ldots, 1 - n_{d-1}/2 \\ 0 \end{array}\right.\right) \quad (13)$$

and

$$p_d^{\text{lin}}(\mathbf{h}_d \,|\, \mathbf{x}) = \frac{\gamma_d}{(2^d\pi\kappa_d^2)^{n_d/2}}G_{0,d}^{d,0}\left(\frac{\|\mathbf{h}_d\|^2}{2^d\kappa_d^2} \left| \begin{array}{c} - \\ 0, (n_1 - n_d)/2, \ldots, (n_{d-1} - n_d)/2 \end{array}\right.\right), \quad (14)$$

respectively, where we define the quantities

$$\kappa_d \equiv \sigma_1 \cdots \sigma_d \|\mathbf{x}\| \quad \text{and} \quad \gamma_d \equiv \prod_{\ell=1}^{d-1} \frac{1}{\Gamma(n_\ell/2)} \quad (15)$$

for brevity. Here, the horizontal dash in the upper row of arguments to $G_{0,d}^{d,0}$ indicates the absence of 'upper' arguments to the $G$-function, denoted by $a_1, \ldots, a_p$ in (12), because $p = 0$. For $d = 2$,

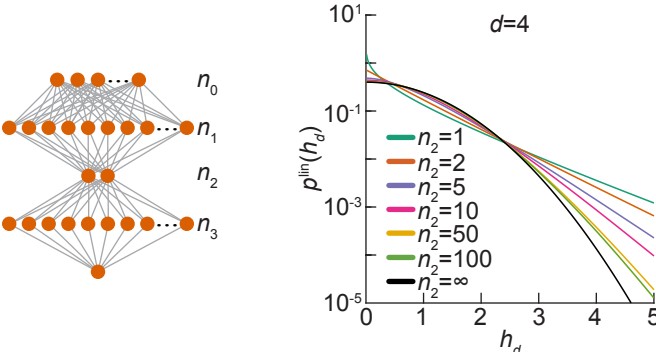

Figure 2: Priors of depth $d = 4$ linear networks with narrow bottlenecks. The left panel shows a diagram of a depth $d = 4$ network with two wide hidden layers of widths $n_1$ and $n_3$ separated by a narrow bottleneck of width $n_2 = 2$. The right panel shows prior densities for networks of this structure with $n_1 = n_3 = 100$ and variable bottleneck widths $n_2$, which is indicated by line color. The prior density is plotted only for positive values of the output preactivation $h_d$, as it is symmetric about zero. The black line indicates the Gaussian limit in which the widths of all three hidden layers are taken to infinity, as discussed in §4.3. Further details on the numerical methods used to generate this figure are provided in Appendix E.

we can use $G$-function identities to recover our earlier results (10) and (11) for a two-layer network (see Appendix A) [29]. We plot the exact density for networks of depths $d = 2, 3$, and $4$ and various widths along with densities estimated from numerical sampling in Figure 1, illustrating that our exact result displays the expected perfect agreement with experiment (see Appendix E for details of our numerical methods).

For any depth, the density (14) has the intriguing property that its functional form depends only on the difference between the hidden layer widths and the output dimensionality. This suggests that the priors of networks with large input and output dimensionalities but narrow intermediate bottlenecks—as would be the case for an autoencoder—will differ noticeably from those of networks with only a few outputs. However, it is challenging to visualize a distribution over more than two variables. We therefore plot the marginal prior over a single component of the output of a network with a bottleneck layer of varying width in Figure 2. Qualitatively, the prior for a network with a narrow bottleneck layer sandwiched between two wide hidden layers is more similar to that of a uniformly narrow network than that of a wide network without a bottleneck. These observations are consistent with previous arguments that wide networks with narrow bottlenecks may possess interesting priors [9, 35].

### 3.3 Deep ReLU networks

Finally, we consider ReLU networks. For this purpose, we adopt a more verbose notation in which the dependence of the prior on width is explicitly indicated, writing $p_d^{\mathrm{lin}}(\mathbf{h}_d; \kappa_d; n_1, \ldots, n_{d-1}, n_d)$ for the prior density (14) of a linear network with the specified hidden layer widths. Similarly, we write $p_d^{\mathrm{ReLU}}(\mathbf{h}_d; \kappa_d; n_1, \ldots, n_{d-1}, n_d)$ for the prior density of the corresponding ReLU network. As shown in Appendix B, we find that

$$
\begin{aligned}
& p_d^{\mathrm{ReLU}}(\mathbf{h}_d; \kappa_d; n_1, \ldots, n_d) \\
& = \left( 1 - \frac{(2^{n_1} - 1)(2^{n_2} - 1) \cdots (2^{n_{d-1}} - 1)}{2^{n_1 + \cdots + n_{d-1}}} \right) \delta(\mathbf{h}_d) \\
& \quad + \frac{1}{2^{n_1 + \cdots + n_{d-1}}} \sum_{k_1=1}^{n_1} \cdots \sum_{k_{d-1}=1}^{n_{d-1}} \binom{n_1}{k_1} \cdots \binom{n_{d-1}}{k_{d-1}} p_d^{\mathrm{lin}}(\mathbf{h}_d; \kappa_d; k_1, \ldots, k_{d-1}, n_d),
\end{aligned} \quad (16)
$$

where $\delta(\mathbf{h}_d)$ is the $n_d$-dimensional Dirac distribution. We prove this result by induction on network depth $d$, using the characteristic function corresponding to this density. The base case $d = 2$ follows by direct integration and the binomial theorem, and the inductive step uses the fact that the linear

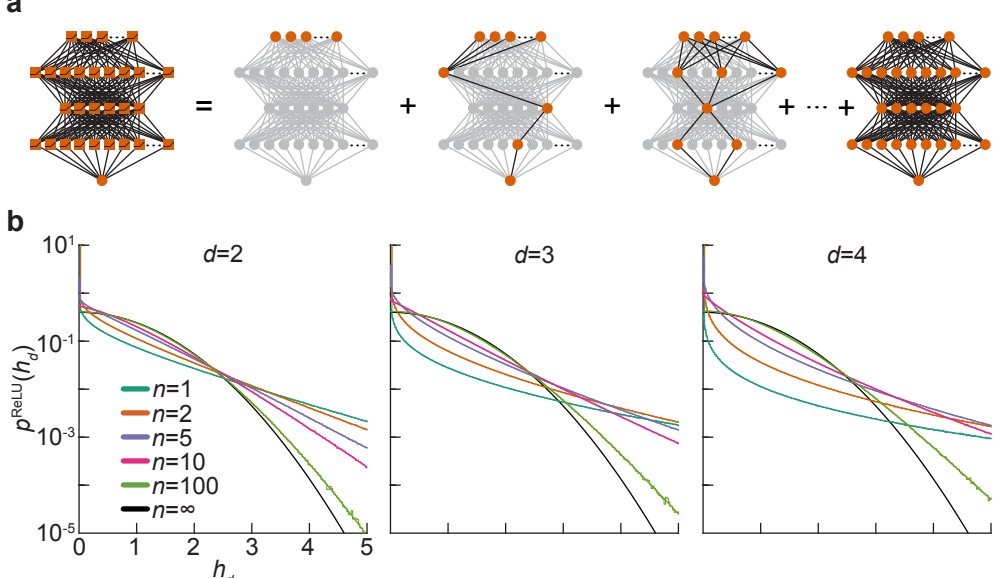

Figure 3: The prior of a deep ReLU network. (**a**) Schematic depiction of the ReLU prior as a mixture of the priors of linear networks of different widths (16). Grey nodes indicate 'inactive' units, while the linear network of active units is shown by the orange nodes. (**b**) ReLU prior densities for networks of depths $d = 2$, 3, and 4 and varying width. Here, we choose $\kappa_d$ such that the variance of the preactivations matches that of the linear networks shown in Figure 1. In each panel, the prior density is plotted only for positive values of the output preactivation $h_d$, as it is symmetric about zero. For each depth, all hidden layers are of the same width $n$, which is indicated by line color. The black line indicates the Gaussian infinite-width limit discussed in §4.3. Thick lines show the exact priors, while thin jagged lines show experimental estimates from $10^8$ examples. Further details on the numerical methods used to generate these figures are provided in Appendix E.

network prior (14) is radial and has marginals equal to the priors of linear networks with fewer outputs. This result has a simple interpretation: the prior for a ReLU network is a mixture of priors of linear networks corresponding to different numbers of active ReLU units in each hidden layer, along with a Dirac distribution representing the cases in which no output units are active. As we did for linear networks, we plot the exact density along with numerical estimates in Figure 3, showing perfect agreement.

# 4 Properties of these priors

Having obtained exact expressions for the priors of deep linear or ReLU networks, we briefly characterize their properties, and how those properties relate to prior analyses of finite network priors.

## 4.1 Moments

We first consider the moments of the output preactivation. As the prior distributions are zero-centered and isotropic, it is clear that all odd raw moments vanish. However, the moments of the norm of the output preactivation are non-vanishing. In particular, using basic properties of the Meijer $G$-function [28, 29], we can easily read off the moments for a linear network as

$$\mathbb{E}_{\text{lin}}\|\mathbf{h}_d\|^m = 2^{dm/2}\kappa_d^m \prod_{\ell=1}^{d}\left(\frac{n_\ell}{2}\right)^{\overline{m/2}} \qquad (m \geq 0), \tag{17}$$

where $a^{\overline{b}} = \Gamma(a+b)/\Gamma(a)$ is the rising factorial [28]. This result takes a particularly simple form for the even moments $m = 2k$, in which case $(n/2)^{\overline{k}} = 2^{-k} \prod_{j=0}^{k-1}(n+2j)$. Most simply, for $m = 2$, we have $\mathbb{E}_{\text{lin}}\|\mathbf{h}_d\|^2 = \kappa_d^2 n_1 \cdots n_d$.

Similarly, for ReLU networks, we have

$$\mathbb{E}_{\text{ReLU}}\|\mathbf{h}_d\|^m = 2^{dm/2}\kappa_d^m \left(\frac{n_d}{2}\right)^{\overline{m/2}} \prod_{\ell=1}^{d-1}\left[\frac{1}{2^{n_\ell}}\sum_{k_\ell=1}^{n_\ell}\binom{n_\ell}{k_\ell}\left(\frac{k_\ell}{2}\right)^{\overline{m/2}}\right]. \tag{18}$$

Each term in the product over $\ell$ expands in terms of generalized hypergeometric functions evaluated at unity [28]. As for linear networks, this expression has a particularly simple form for even moments, particularly if $m = 2$, for which $\mathbb{E}_{\text{ReLU}}\|\mathbf{h}_d\|^2 = 2^{1-d}\kappa_d^2 n_1 \cdots n_d$. Therefore, for identical weight variances, the variance of the output preactivation of a ReLU network is $2^{1-d}$ times that of a linear network of the same width and depth. However, one can compensate for this variance reduction by simply doubling the variances of the priors over the hidden layer weights.

Using the property that the marginal prior distribution of a single component of the output is identical to the prior of a single-output network, these results give the marginal absolute moments of the prior of a linear or ReLU network. Moreover, these results can also be used to obtain joint moments of different components by exploiting the fact that the prior is radial. By symmetry, the odd moments vanish, and the even moments are given up to combinatorial factors by the corresponding moments of any individual component of the preactivation. For example, the covariance of two components of the output preactivation is $\mathbb{E}h_{d,i}h_{d,j} = (\mathbb{E}h_{d,1}^2)\delta_{ij}$ for all $i, j = 1, \ldots, n_d$.

## 4.2 Tail bounds

Vladimirova et al. [21, 22] have shown that the marginal prior distributions of the preactivations of deep networks with ReLU-like activation functions and fixed, finite widths become increasingly heavy-tailed with depth. This behavior contrasts sharply with the thin-tailed Gaussian prior of infinite-width networks [4–8]. In particular, Vladimirova et al. [21, 22] showed that the prior distributions are sub-Weibull with optimal tail parameter $\theta = d/2$, meaning that they satisfy

$$\mathbb{P}(|h_{d,j}| \geq \rho) \leq C \exp(-\rho^{1/\theta}) \tag{19}$$

for each neuron $j \in \{1, \ldots, n_d\}$, all $\rho > 0$, and some constant $C > 0$ if $\theta \geq d/2$, but not if $\theta < d/2$. A sub-Gaussian distribution is sub-Weibull with optimal tail parameter at most $1/2$; distributions with larger tail parameters have increasingly heavy tails. As shown in Appendix C, we can use the results of §4.1 to give a straightforward derivation of this result, showing that the norm $\|\mathbf{h}_d\|$ of the output preactivation for either linear or ReLU networks is sub-Weibull with optimal tail parameter $d/2$. Due to the aforementioned fact that the marginal prior for a single output of a multi-output network is identical to the prior for a single-output network, this implies (19).

## 4.3 Asymptotic behavior

Most previous studies of the priors of deep Bayesian networks have focused on their asymptotic behavior for large hidden layer widths. Provided that one takes

$$\kappa_d = (n_1 \cdots n_{d-1})^{-1/2} \varkappa_d \tag{20}$$

for $\varkappa_d$ independent of the hidden layer widths such that the preactivation variance remains finite, the prior tends to a Gaussian as $n_1, \cdots, n_{d-1} \to \infty$ for fixed $d$, $n_0$, and $n_d$ [4–10, 12, 24]. This behavior is qualitatively apparent in Figures 1 and 3. Here, we exploit our exact results to study this asymptotic regime. An ideal approach would be to study the asymptotic behavior of the characteristic function (13) and apply Lévy's continuity theorem [36] to obtain the Gaussian limit, but we are not aware of suitable doubly-scaled asymptotic expansions for the Meijer $G$-function [28, 29]. Instead, we use a multivariate Edgeworth series to obtain an asymptotic expansion of the density [37]. As

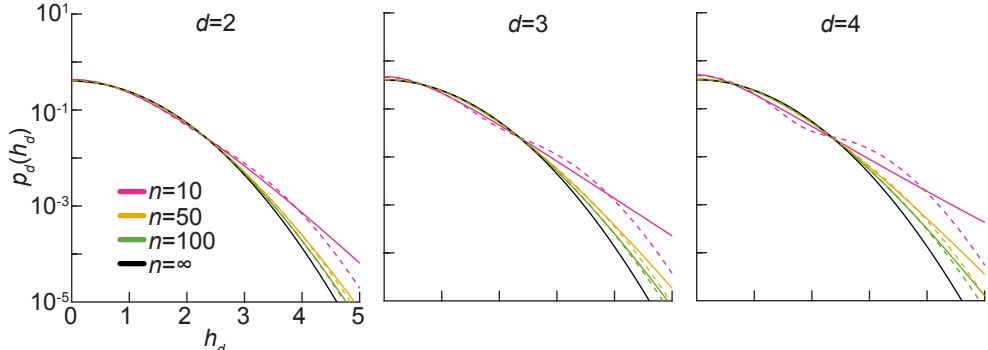

Figure 4: The large-width Edgeworth approximation for the prior density is thin-tailed. From left to right, the panels show the priors of linear networks of depths $d = 2, 3$, and $4$ of varying widths. In each panel, solid lines show the exact prior density (14), while dashed lines show the asymptotic Edgeworth approximation (21). The exact prior density is computed numerically as described in Appendix E.

detailed in Appendix D, we find that the prior of a linear network has an Edgeworth series of the form

$$p_d^{\text{lin}}(\mathbf{h}_d \mid \mathbf{x}) \approx \frac{1}{(2\pi \varkappa_d^2)^{n_d/2}} \exp\left(-\frac{\|\mathbf{h}_d\|^2}{2\varkappa_d^2}\right)$$
$$\times \left[1 + \frac{1}{4}\left(\sum_{\ell=1}^{d-1} \frac{1}{n_\ell}\right)\left(\frac{\|\mathbf{h}_d\|^4}{\varkappa_d^4} - 2(n_d + 2)\frac{\|\mathbf{h}_d\|^2}{\varkappa_d^2} + n_d(n_d + 2)\right) + \mathcal{O}\left(\frac{1}{n^2}\right)\right].$$
$$(21)$$

The Edgeworth expansion of the prior of a ReLU network is of the same form, with the factor of $1/4$ scaling the finite-width correction being replaced by $5/4$, and the variance $\varkappa_d^2$ re-scaled by $2^{1-d}$. Heuristically, this result makes sense given that the binomial sums in (16) will be dominated by $k_\ell \approx n_\ell/2$ in the large-width limit.

These results succinctly reproduce the leading finite-width corrections formally written down by Antognini [15] and recursively computed by Yaida [12]. However, importantly, this approximate distribution is sub-Gaussian: it cannot capture the depth-dependent heaviness of the tails of the true finite-width prior described in §4.2. More generally, one can see that the heavier-than-Gaussian tails of the finite width prior are an essentially non-perturbative effect. At any finite order of the Edgeworth expansion, the approximate density for a network of any fixed depth is of the form $(2\pi \varkappa_d^2)^{-n_d/2} \exp(-\|\mathbf{h}_d\|^2/2\varkappa_d^2)[1 + f(\|\mathbf{h}_d\|^2/\varkappa_d^2)]$, where $f$ is a polynomial satisfying $\int d\mathbf{h} \, \exp(-\|\mathbf{h}\|^2/2)f(\|\mathbf{h}\|^2) = 0$ [37]. Such a density is sub-Gaussian. In Figure 4, we illustrate the discrepancy between the thin tails of the Edgeworth expansion and the heavier tails of the exact prior. Even at the relatively modest depths shown, the increasing discrepancy between the tail behavior of the approximate prior and the true tail behavior with increasing depth is clearly visible. We emphasize that low-order Edgeworth expansions will capture some qualitative features of the finite-width prior, but not all. It is therefore important to consider approximation accuracy on a case-by-case basis depending on what features of finite BNNs one aims to study.

## 5   Related work

As previously mentioned, our work closely relates to a program that proposes to study finite BNNs perturbatively by calculating asymptotic corrections to the prior [12, 14]. Though these approaches are applicable to the prior over outputs for multiple input examples and to more general activation functions, they are valid only in the regime of large hidden layer widths. As detailed in §4.3, these asymptotic results can be obtained as a limiting case of our exact solutions, though the Edgeworth series does not capture the heavier-than-Gaussian tails of the true finite-width prior. In a similar vein, recent works have perturbatively studied the finite-width posterior for a Gaussian likelihood

[12, 13, 38]. Our work is particularly similar in spirit to that of Schoenholz et al. [13], who considered asymptotic approximations to the partition function of the single-example function space posterior. Our exact solutions for simple models provide a broadly useful point of comparison for future perturbative study [12–15, 25–27].

As discussed in §4.2, our exact results recapitulate the observation of Vladimirova et al. [21, 22] that the prior distributions of finite networks become increasingly heavy-tailed with depth. Moreover, our results are consistent with the work of Gur-Ari and colleagues, who showed that the moments we compute should remain bounded at large widths [16, 17]. Similar results on the tail behavior of deep Gaussian processes, of which BNNs are a degenerate subclass, have recently been obtained by Lu et al. [39] and by Pleiss and Cunningham [40]. Our approach complements the study of tail bounds and asymptotic moments. Exact solutions provide a finer-grained characterization of the prior, but it is possible to compute tail bounds and moments for models for which the exact prior is not straightforwardly calculable. We note that, following the appearance of our work in preprint form and after the submission deadline, parallel results on exact marginal function-space priors were announced by Noci et al. [41].

After the completion of our work, we became aware of the close connection of our results on deep linear network priors to previous work in random matrix theory (RMT). In the language of RMT, the marginal function-space prior of a deep linear BNN is a particular linear statistic of the product of rectangular real Ginibre matrices (i.e., matrices with independent and identically distributed Gaussian entries) [42]. An alternative proof of the result (14) then follows by using the rotational invariance of what is known in RMT as the one-point weight function of the singular value distribution of the product matrix [43]. However, to the best of our knowledge, this connection had not previously been exploited to study the properties of finite linear BNNs. Further non-asymptotic study of random matrix products, and of nonlinear compositions as in the deep ReLU BNNs considered here, will be an interesting objective for future work [42–44].

Finally, previous works have theoretically and empirically investigated how finite-width network priors affect inference [9–11, 18–20, 23]. Some of these studies observed an intriguing phenomenon: better generalization performance is obtained when inference is performed using a "cold" posterior that is artificially tempered as $p(\mathbf{h}_d \mid \mathbf{x}, \mathbf{y})^{1/T}$ for $0 < T < 1$ [11, 18, 19]. This contravenes the expectation that the Bayes posterior (i.e., $T = 1$) should be optimal. It has been suggested that this effect reflects misspecification either of the prior over the weights—namely, that a distribution other than an isotropic Gaussian should be employed —or of the likelihood [11], but the true cause remains unclear [20]. The exact function space priors computed in this work should prove useful in ongoing dissections of simple models for BNN inference. Most simply, they provide a finer-grained understanding of how hyperparameter choice affects the prior than that afforded by tail bounds alone [21, 22, 40]. Though we do not compute function-space posterior distributions, knowing the precise form of the prior would allow one to gain an intuitive understanding of the shape of the posterior for a given likelihood [10]. For instance, one could imagine a particular degree of heavy-tailedness in the prior being optimal for a dataset that is to some degree heavy-tailed. This could allow one to gain some intuition for when the prior or likelihood is misspecified for a given dataset. Detailed experimental and analytical investigation of these questions is an important objective of our future work.

# 6   Conclusions

In this paper, we have performed the first exact characterization of the function-space priors of finite deep Bayesian neural networks induced by Gaussian priors over their weights. These exact solutions provide a useful check on the validity of perturbative studies [12, 14, 15], and unify previous descriptions of finite-width network priors [12, 14–17, 21, 22]. Our solutions were, however, obtained for the relatively restrictive setting of the marginal prior for a single input example of a feedforward network with no bias terms. As our approach relies heavily on rotational invariance, it is unclear how best to generalize these methods to networks with non-zero bias terms, or to the joint prior of the output preactivations for multiple inputs. We therefore leave detailed study of those general settings as an interesting objective for future work.

## Acknowledgments and Disclosure of Funding

We thank A. Atanasov, B. Bordelon, A. Canatar, and M. Farrell for helpful comments on our manuscript. The computations in this paper were performed using the Harvard University FAS Division of Science Research Computing Group's Cannon HPC cluster. JAZ-V acknowledges support from the NSF-Simons Center for Mathematical and Statistical Analysis of Biology at Harvard and the Harvard Quantitative Biology Initiative. CP thanks Intel, Google, and the Harvard Data Science Initiative for support. The authors declare no competing interests.

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
