# Supplemental Material for: "Exact marginal prior distributions of finite Bayesian neural networks"

**Jacob A. Zavatone-Veth**
Department of Physics
Harvard University
Cambridge, MA 02138
jzavatoneveth@g.harvard.edu

**Cengiz Pehlevan**
John A. Paulson School of Engineering and Applied Sciences
Harvard University
Cambridge, MA 02138
cpehlevan@seas.harvard.edu

## Contents


# A   Derivation of the prior of a deep linear network

In this appendix, we prove the formula for the prior of a deep linear network given in §3 of the main text. In §A.1, we prove that the claimed density and characteristic function are indeed a Fourier transform pair using identities for the Hankel transform, and then prove by induction that these results describe the prior of a deep linear network in §A.2. Finally, we provide a lengthier, albeit possibly more transparent, proof of these results by direct integration in §A.3.

## A.1   Fourier transforms of radial functions and the Hankel transform

We begin by reviewing the relationship between the Fourier transform of a radial function and the Hankel transform, and then use this relationship to prove that the claimed characteristic function and density are a Fourier transform pair. Let $p, \varphi : \mathbb{R}^n \to \mathbb{R}$ be a Fourier transform pair, with

$$\varphi(\mathbf{q}) = \int d\mathbf{h} \, \exp(-i\mathbf{h} \cdot \mathbf{q}) p(\mathbf{h}) \quad \text{and} \quad p(\mathbf{h}) = \int \frac{d\mathbf{q}}{(2\pi)^n} \exp(i\mathbf{h} \cdot \mathbf{q}) \varphi(\mathbf{q}). \tag{A.1}$$

Assume that $p$ and $\varphi$ are radial functions, i.e., that $p(\mathbf{h}) = p(\|\mathbf{h}\|)$ and $\varphi(\mathbf{q}) = \varphi(\|\mathbf{q}\|)$. We note that if one of $p$ or $\varphi$ is radial, it follows that both are radial [1]. Then, we have the Hankel transform relations

$$\varphi(\mathbf{q}) = (2\pi)^{+n/2} \|\mathbf{q}\|^{(2-n)/2} \int_0^\infty r \, dr \, J_{(n-2)/2}(\|\mathbf{q}\|r) r^{(n-2)/2} p(r) \tag{A.2}$$

$$p(\mathbf{h}) = (2\pi)^{-n/2} \|\mathbf{h}\|^{(2-n)/2} \int_0^\infty r \, dr \, J_{(n-2)/2}(\|\mathbf{h}\|r) r^{(n-2)/2} \varphi(r), \tag{A.3}$$

where $J_\nu(z)$ is the Bessel function of the first kind of order $\nu$ [2, 1, 3–5]. We note that inversion of the Hankel transform formally follows from the distributional identity

$$\int_0^\infty r \, dr \, J_\nu(kr) J_\nu(k'r) = \frac{\delta(k - k')}{k} \tag{A.4}$$

for $k, k' > 0$ [2, 1, 3–5].

We now use this relationship to show that

$$p_d^{\mathrm{lin}}(\mathbf{h}_d \,|\, \mathbf{x}) = \frac{\gamma_d}{(2^d \pi \kappa_d^2)^{n_d/2}} G_{0,d}^{d,0} \left( \frac{\|\mathbf{h}_d\|^2}{2^d \kappa_d^2} \,\middle|\, \begin{matrix} - \\ 0, (n_1 - n_d)/2, \ldots, (n_{d-1} - n_d)/2 \end{matrix} \right) \tag{A.5}$$

and

$$\varphi_d^{\mathrm{lin}}(\mathbf{q}_d \,|\, \mathbf{x}) = \gamma_d G_{d-1,1}^{1,d-1} \left( 2^{d-2} \kappa_d^2 \|\mathbf{q}_d\|^2 \,\middle|\, \begin{matrix} 1 - n_1/2, \ldots, 1 - n_{d-1}/2 \\ 0 \end{matrix} \right) \tag{A.6}$$

are a Fourier transform pair, where $\kappa_d, \gamma_d > 0$ and $n_1, \ldots, n_d \in \mathbb{N}_{>0}$. As both of these $G$-functions are well-behaved, it suffices to show one direction of this relationship; we will show that $p_d$ is the inverse Fourier transform of $\varphi_d$. Our starting point is the formula for the Hankel transform of a $G$-function multiplied by a power:

$$\int_0^\infty dx \, J_\nu(xy) x^{2\rho} G_{p,q}^{m,n} \left( \lambda x^2; \begin{matrix} a_1, \ldots, a_p \\ b_1, \ldots, b_q \end{matrix} \right) = \frac{2^{2\rho}}{y^{2\rho+1}} G_{p+2,q}^{m,n+1} \left( \frac{4\lambda}{y^2}; \begin{matrix} h, a_1, \ldots, a_p, k \\ b_1, \ldots, b_q \end{matrix} \right) \tag{A.7}$$

where $h = 1/2 - \rho - \nu/2$ and $k = 1/2 - \rho + \nu/2$, valid for $p + q < 2(m + n)$, all real $\lambda$, $\Re(b_j + \rho + \nu/2) > -1/2$, and $\Re(a_j + \rho) < 3/4$ [5]. Using this identity and simplifying the result using the $G$-function identities [2, 3]

$$G_{p,q}^{m,n} \left( \frac{1}{z} \,\middle|\, \begin{matrix} a_1, \ldots, a_p \\ b_1, \ldots, b_q \end{matrix} \right) = G_{q,p}^{n,m} \left( z \,\middle|\, \begin{matrix} 1 - b_1, \ldots, 1 - b_q \\ 1 - a_1, \ldots, 1 - a_p \end{matrix} \right) \tag{A.8}$$

and

$$z^\mu G_{p,q}^{m,n} \left( z \,\middle|\, \begin{matrix} a_1, \ldots, a_p \\ b_1, \ldots, b_q \end{matrix} \right) = G_{p,q}^{m,n} \left( z \,\middle|\, \begin{matrix} a_1 + \mu, \ldots, a_p + \mu \\ b_1 + \mu, \ldots, b_q + \mu \end{matrix} \right), \tag{A.9}$$

we obtain

$$p_d^{\text{lin}}(\mathbf{h}_d \,|\, \mathbf{x}) = \frac{\gamma_d}{(2^d \kappa_d^2)^{n_d/2}} G_{1,d+1}^{d,1} \left( \frac{\|\mathbf{h}_d\|^2}{2^d \kappa_d^2} \,\middle|\, \begin{matrix} 1 - n_d/2 \\ 0, (n_1 - n_d)/2, \dots, (n_{d-1} - n_d)/2, 1 - n_d/2 \end{matrix} \right).$$

(A.10)

Then, further simplifying using the identity [2, 3]

$$G_{p+1,q+1}^{m,n+1} \left( z \,\middle|\, \begin{matrix} \alpha, a_1, \dots, a_p \\ b_1, \dots, b_q, \alpha \end{matrix} \right) = G_{p,q}^{m,n} \left( z \,\middle|\, \begin{matrix} a_1, \dots, a_p \\ b_1, \dots, b_q \end{matrix} \right),$$

(A.11)

we conclude the desired result. The proof that $\varphi_d$ is the Fourier transform of $p_d$ can be derived by an analogous procedure.

## A.2   Inductive proof of the $G$-function formula

We now prove the claimed formula for the prior by induction on the depth $d$. Using the identities [3]

$$G_{0,2}^{2,0} \left( z \,\middle|\, \begin{matrix} - \\ 0, (n_1 - n_2)/2 \end{matrix} \right) = 2z^{(n_1 - n_2)/4} K_{(n_1 - n_2)/2}(2\sqrt{z})$$

(A.12)

and

$$G_{1,1}^{1,1} \left( z \,\middle|\, \begin{matrix} 1 - n_1/2 \\ 0 \end{matrix} \right) = \Gamma \left( \frac{n_1}{2} \right) (1 + z)^{-n_1/2},$$

(A.13)

the claim for the density and characteristic function for the base case $d = 2$ follow from the direct calculation in §3 of the main text, specifically equations (10) and (11).

For $d > 2$, we observe that the general formula for the characteristic function (9) implies the recursive integral relation

$$\varphi_{d+1}^{\text{lin}}(\mathbf{q}_{d+1} \,|\, \mathbf{x}) = \int d\mathbf{h}_d \, \exp \left( -\frac{1}{2} \sigma_{d+1}^2 \|\mathbf{h}_d\|^2 \|\mathbf{q}_{d+1}\|^2 \right) p_d(\mathbf{h}_d \,|\, \mathbf{x}).$$

(A.14)

On the induction hypothesis, this yields

$$\varphi_{d+1}^{\text{lin}}(\mathbf{q}_{d+1} \,|\, \mathbf{x}) = \frac{\gamma_d}{(2^d \pi \kappa_d^2)^{n_d/2}} \int d\mathbf{h}_d \, \exp \left( -\frac{1}{2} \sigma_{d+1}^2 \|\mathbf{h}_d\|^2 \|\mathbf{q}_{d+1}\|^2 \right)$$
$$\times G_{0,d}^{d,0} \left( \frac{\|\mathbf{h}_d\|^2}{2^d \kappa_d^2} \,\middle|\, \begin{matrix} - \\ 0, \nu_1, \dots, \nu_{d-1} \end{matrix} \right),$$

(A.15)

where we define $\nu_\ell \equiv (n_\ell - n_d)/2$ for $\ell = 1, \dots, d - 1$ for brevity. Converting to spherical coordinates and evaluating the trivial angular integral, we have

$$\varphi_{d+1}^{\text{lin}}(\mathbf{q}_{d+1} \,|\, \mathbf{x}) = \gamma_{d+1} \int_0^\infty dt \, t^{n_d/2 - 1} \exp \left( -2^{d-1} \kappa_{d+1}^2 \|\mathbf{q}_{d+1}\|^2 t \right) G_{0,d}^{d,0} \left( t \,\middle|\, \begin{matrix} - \\ 0, \nu_1, \dots, \nu_{d-1} \end{matrix} \right),$$

(A.16)

where we have made the change of variables $t \equiv h_d^2/2^d \kappa_d^2$ and recognized $\kappa_{d+1} = \sigma_{d+1} \kappa_d$ and $\gamma_{d+1} = \gamma_d / \Gamma(n_d/2)$. We now recall the formula for the Laplace transform of a $G$-function multiplied by a power:

$$\int_0^\infty dt \, \exp(-zt) t^{-\alpha} G_{p,q}^{m,n} \left( t \,\middle|\, \begin{matrix} a_1, \dots, a_p \\ b_1, \dots, b_q \end{matrix} \right) = z^{\alpha-1} G_{p+1,q}^{m,n+1} \left( \frac{1}{z} \,\middle|\, \begin{matrix} \alpha, a_1, \dots, a_p \\ b_1, \dots, b_q \end{matrix} \right),$$

(A.17)

valid either if $p + q < 2(m + n)$ and $\Re(\alpha) > \Re(b_j + 1)$ for all $j = 1, \dots, m$ or if $p < q$ and $\Re(\alpha) < \Re(b_j + 1)$ for all $j = 1, \dots, m$, and for $|\arg z| < (m + n - p/2 - q/2)\pi$ [4]. The latter condition applies, hence, using the identity (A.8), we find that

$$\varphi_{d+1}^{\text{lin}}(\mathbf{q}_{d+1} \,|\, \mathbf{x}) = \gamma_{d+1} (2^{d-1} \kappa_{d+1}^2 \|\mathbf{q}_{d+1}\|^2)^{-n_d/2}$$
$$\times G_{d,1}^{1,d} \left( 2^{d-1} \kappa_{d+1}^2 \|\mathbf{q}_{d+1}\|^2 \,\middle|\, \begin{matrix} 1, 1 - \nu_1, \dots, 1 - \nu_{d-1} \\ n_d/2 \end{matrix} \right).$$

(A.18)

Then, applying the identity (A.9), we obtain

$$\varphi_{d+1}^{\text{lin}}(\mathbf{q}_{d+1} \,|\, \mathbf{x}) = \gamma_{d+1} G_{d,1}^{1,d} \left( 2^{d-1} \kappa_{d+1}^2 \|\mathbf{q}_{d+1}\|^2 \,\middle|\, \begin{matrix} 1 - n_1/2, \dots, 1 - n_d/2 \\ 0 \end{matrix} \right),$$

(A.19)

where we have used the fact that the $G$-function is invariant under permutation of its upper arguments. Therefore, using the results of §A.1, we conclude the claimed result.

## A.3 Derivation of the prior by direct integration

Here, we directly derive a formula for the prior as a $(d-1)$-dimensional integral, and then show that this is equivalent to the expression in terms of the Meijer $G$-function. Separating out the terms that correspond to the first and last layers, the general expression for the characteristic function (9) becomes

$$\varphi_d^{\text{lin}}(\mathbf{q}_d) = \int \prod_{\ell=1}^{d-1} \frac{d\mathbf{q}_\ell \, d\mathbf{h}_\ell}{(2\pi)^{n_\ell}} \exp\left( \sum_{\ell=1}^{d-1} i\mathbf{q}_\ell \cdot \mathbf{h}_\ell - \frac{1}{2}\sigma_1^2 \|\mathbf{x}\|^2 \|\mathbf{q}_1\|^2 \right.$$
$$\left. - \frac{1}{2}\sum_{\ell=2}^{d-1} \sigma_\ell^2 \|\mathbf{q}_\ell\|^2 \|\mathbf{h}_{\ell-1}\|^2 - \frac{1}{2}\sigma_d^2 \|\mathbf{q}_d\|^2 \|\mathbf{h}_{d-1}\|^2 \right), \quad \text{(A.20)}$$

where we suppress the fact that $\varphi_d$ is implicitly conditioned on $\mathbf{x}$. Transforming into spherical coordinates and evaluating the angular integrals as described in Appendix A.1, we obtain

$$\varphi_d^{\text{lin}}(\mathbf{q}_d) = \left[ \prod_{\ell=1}^{d-1} \frac{2^{1-n_\ell/2}}{\Gamma(n_\ell/2)} \right] \left[ \prod_{\ell=1}^{d-1} \int_0^\infty dh_\ell \int_0^\infty dq_\ell \, (h_\ell q_\ell)^{n_\ell/2} J_{(n_\ell-2)/2}(h_\ell q_\ell) \right]$$
$$\times \exp\left( -\frac{1}{2}\sigma_1^2 \|\mathbf{x}\|^2 q_1^2 - \frac{1}{2}\sum_{\ell=2}^{d-1} \sigma_\ell^2 q_\ell^2 h_{\ell-1}^2 - \frac{1}{2}\sigma_d^2 \|\mathbf{q}_d\|^2 h_{d-1}^2 \right). \quad \text{(A.21)}$$

Assuming that $\sigma_\ell > 0$ and $\mathbf{x} \neq 0$, we make the change of variables

$$u_\ell \equiv \sigma_\ell \sigma_{\ell-1} \cdots \sigma_1 \|\mathbf{x}\| q_\ell \quad \text{(A.22)}$$

$$v_\ell \equiv \frac{1}{\sigma_\ell \sigma_{\ell-1} \cdots \sigma_1} \frac{1}{\|\mathbf{x}\|} h_\ell \quad \text{(A.23)}$$

such that

$$\sigma_\ell^2 = u_\ell^2 v_{\ell-1}^2 \quad \text{(A.24)}$$

and

$$q_\ell h_\ell = u_\ell v_\ell. \quad \text{(A.25)}$$

This yields

$$\varphi_d^{\text{lin}}(\mathbf{q}_d) = \left[ \prod_{\ell=1}^{d-1} \frac{2^{1-n_\ell/2}}{\Gamma(n_\ell/2)} \right] \left[ \prod_{\ell=1}^{d-1} \int_0^\infty dv_\ell \int_0^\infty du_\ell \, (v_\ell u_\ell)^{n_\ell/2} J_{(n_\ell-2)/2}(v_\ell u_\ell) \right]$$
$$\times \exp\left( -\frac{1}{2}u_1^2 - \frac{1}{2}\sum_{\ell=2}^{d-1} u_\ell^2 v_{\ell-1}^2 - \frac{1}{2}\kappa_d^2 v_{d-1}^2 \|\mathbf{q}_d\|^2 \right), \quad \text{(A.26)}$$

where we write

$$\kappa_d \equiv \sigma_d \sigma_{d-1} \cdots \sigma_1 \|\mathbf{x}\| \quad \text{(A.27)}$$

for brevity.

At this stage, we shift to considering the prior density, following the results of §A.1. Using the identity [2, 6]

$$\int_0^\infty du_\ell \, u_\ell^{n_\ell/2} J_{(n_\ell-2)/2}(v_\ell u_\ell) \exp\left( -\frac{1}{2}v_{\ell-1}^2 u_\ell^2 \right) = v_\ell^{n_\ell/2-1} v_{\ell-1}^{-n_\ell} \exp\left( -\frac{1}{2}\frac{v_\ell^2}{v_{\ell-1}^2} \right) \quad \text{(A.28)}$$

to integrate out the variables $u_\ell$ and $q_d$, we obtain

$$p_d^{\text{lin}}(\mathbf{h}_d) = \frac{\kappa_d^{-n_d}}{(2\pi)^{n_d/2}} \left[ \prod_{\ell=1}^{d-1} \frac{2^{1-n_\ell/2}}{\Gamma(n_\ell/2)} \right] \left[ \prod_{\ell=1}^{d-1} \int_0^\infty dv_\ell \, v_\ell^{n_\ell-n_{\ell+1}-1} \right]$$
$$\times \exp\left( -\frac{1}{2}v_1^2 - \frac{1}{2}\sum_{\ell=2}^{d-1} \frac{v_\ell^2}{v_{\ell-1}^2} - \frac{1}{2}\frac{\|\mathbf{h}_d\|^2}{\kappa_d^2 v_{d-1}^2} \right). \quad \text{(A.29)}$$

S5

We now make a change of variables to decouple all but one of the terms in the exponential. In particular, we let

$$s_\ell \equiv \begin{cases} v_1 & \ell = 1 \\ v_\ell/v_{\ell-1} & 1 < \ell \leq d-1, \end{cases} \tag{A.30}$$

such that

$$v_\ell = s_\ell s_{\ell-1} \cdots s_1. \tag{A.31}$$

The Jacobian of this transformation is lower triangular, and can be seen to have determinant

$$\left| \det \frac{\partial(v_1, \ldots, v_{d-1})}{\partial(s_1, \ldots, s_{d-1})} \right| = \frac{1}{s_1 s_2 \cdots s_{d-1}} \prod_{\ell=1}^{d-1} v_\ell, \tag{A.32}$$

which is non-singular on all but a measure-zero subset of the integration domain. This yields

$$\left| \det \frac{\partial(v_1, \ldots, v_{d-1})}{\partial(s_1, \ldots, s_{d-1})} \right| \prod_{\ell=1}^{d-1} v_\ell^{n_\ell - n_{\ell+1} - 1} = \frac{1}{s_1 s_2 \cdots s_{d-1}} \prod_{\ell=1}^{d-1} v_\ell^{n_\ell - n_{\ell+1}} = \prod_{\ell=1}^{d-1} s_\ell^{n_\ell - n_d - 1}, \tag{A.33}$$

hence the prior density becomes

$$p_d^{\text{lin}}(\mathbf{h}_d) = \frac{\kappa_d^{-n_d}}{(2\pi)^{n_d/2}} \left[ \prod_{\ell=1}^{d-1} \frac{2^{1-n_\ell/2}}{\Gamma(n_\ell/2)} \right] \left[ \prod_{\ell=1}^{d-1} \int_0^\infty ds_\ell \, s_\ell^{n_\ell - n_d - 1} \exp(-s_\ell^2/2) \right]$$
$$\times \exp\left( -\frac{1}{2} \frac{\|\mathbf{h}_d\|^2}{\kappa_d^2} \frac{1}{s_1^2 s_2^2 \cdots s_{d-1}^2} \right). \tag{A.34}$$

For convenience, we make a final change of variables

$$t_\ell \equiv \frac{1}{2} s_\ell^2, \tag{A.35}$$

which yields the formula

$$p_d^{\text{lin}}(\mathbf{h}_d \mid \mathbf{x}) = \frac{\gamma_d}{(2^d \pi \kappa_d^2)^{n_d/2}} f_{d-1}\left( \frac{\|\mathbf{h}_d\|^2}{2^d \kappa_d^2}; \frac{n_1 - n_d}{2}, \ldots, \frac{n_{d-1} - n_d}{2} \right), \tag{A.36}$$

where we define

$$\gamma_d \equiv \prod_{\ell=1}^{d-1} \frac{1}{\Gamma(n_\ell/2)} \tag{A.37}$$

as in the main text, as well as the integral function

$$f_q(z; \nu_1, \ldots, \nu_q) \equiv \left[ \prod_{j=1}^q \int_0^\infty dt_j \, t_j^{\nu_j - 1} \exp(-t_j) \right] \exp\left( -z \frac{1}{t_1 \cdots t_q} \right) \tag{A.38}$$

for parameters $\nu_j \in \mathbb{R}$ and $z \geq 0$. The claim is that

$$f_q(z; \nu_1, \ldots, \nu_q) = G_{0,q+1}^{q+1,0}\left( z \left| \begin{matrix} - \\ 0, \nu_1, \ldots, \nu_q \end{matrix} \right. \right), \tag{A.39}$$

which follows directly from the Mellin transform $\mathcal{M}f_q$ of $f_q$ and the definition of the Meijer $G$-function as the Mellin-Barnes integral (12). For $s \in \mathbb{C}$ such that $\Re(s) > 0$ and $\Re(\nu_j + s) > 0$ for all $j$, we can easily compute [4]

$$\{\mathcal{M}f_q\}(s; \nu_1, \ldots, \nu_q) = \int_0^\infty dz \, z^{s-1} f_q(s; \nu_1, \ldots, \nu_q) = \Gamma(s) \prod_{j=1}^q \Gamma(\nu_j + s). \tag{A.40}$$

For $s$ satisfying the above properties, the properties of the $\Gamma$ function imply that $\mathcal{M}f_q$ is a function that tends to zero uniformly as $\Im(s) \to \pm\infty$. Then, by the Mellin inversion theorem [2, 4], we have

$$f_q(z; \nu_1, \ldots, \nu_q) = \frac{1}{2\pi i} \int_{c-i\infty}^{c+i\infty} ds \, z^{-s} \Gamma(s) \prod_{j=1}^q \Gamma(\nu_j + s) \tag{A.41}$$

where the contour is chosen such that $\Re(s) = c$ satisfies the above conditions. This is the definition of the desired Meijer $G$-function [2, 3], hence we conclude the claimed result.

# B Derivation of the prior of a deep ReLU network

In this appendix, we derive the expansion given in §3.3 for the prior of a ReLU network as a mixture of the priors of linear networks of varying widths. Using the linearity of the Fourier transform, the desired result can be stated in terms of characteristic functions as

$$
\begin{aligned}
\varphi_d^{\text{ReLU}}&(\mathbf{q}_d; \kappa_d; n_1, \ldots, n_{d-1}) \\
&= 1 - \frac{(2^{n_1} - 1)(2^{n_2} - 1) \cdots (2^{n_{d-1}} - 1)}{2^{n_1 + \cdots + n_{d-1}}} \\
&\quad + \frac{1}{2^{n_1 + \cdots + n_{d-1}}} \sum_{k_1=1}^{n_1} \cdots \sum_{k_{d-1}=1}^{n_{d-1}} \binom{n_1}{k_1} \cdots \binom{n_{d-1}}{k_{d-1}} \varphi_d^{\text{lin}}(\mathbf{q}_d; \kappa_d; k_1, \ldots, k_{d-1}).
\end{aligned}
\tag{B.1}
$$

We prove this proposition by induction on the depth $d$.

For a network with a single hidden layer, we can easily evaluate the characteristic function $\varphi_2$ for $\phi_1(x) = \max\{0, x\}$ as the integrals factor over the hidden layer dimensions, yielding

$$
\varphi_2^{\text{ReLU}}(\mathbf{q}_2) = \left[ \frac{1}{2} + \frac{1}{2} \left( 1 + \kappa_2^2 \|\mathbf{q}_2\|^2 \right)^{-1/2} \right]^{n_1},
\tag{B.2}
$$

where, as before, $\kappa_2 \equiv \sigma_1 \sigma_2 \|\mathbf{x}\|$. Expanding this result using the binomial theorem, we find that

$$
\begin{aligned}
\varphi_2^{\text{ReLU}}(\mathbf{q}_2; \kappa_2; n_1) &= \frac{1}{2^{n_1}} \sum_{k=0}^{n_1} \binom{n_1}{k} \left( 1 + \kappa_2^2 \|\mathbf{q}_2\|^2 \right)^{-k/2} \\
&= \frac{1}{2^{n_1}} + \frac{1}{2^{n_1}} \sum_{k=1}^{n_1} \binom{n_1}{k} \varphi_2^{\text{lin}}(\mathbf{q}_2; \kappa_2; k),
\end{aligned}
\tag{B.3}
$$

which proves the base case of the desired result.

We now consider a depth $d$ network. From the definition of the characteristic functions, we have the recursive identity

$$
\begin{aligned}
\varphi_d^{\text{ReLU}}(\mathbf{q}_d; \kappa_d; n_1, \ldots, n_{d-1}) = \int \frac{d\mathbf{q}_{d-1} \, d\mathbf{h}_{d-1}}{(2\pi)^{n_{d-1}}} \exp\left( i\mathbf{q}_{d-1} \cdot \mathbf{h}_{d-1} - \frac{1}{2} \sigma_d^2 \|\mathbf{q}_d\|^2 \|\phi(\mathbf{h}_{d-1})\|^2 \right) \\
\times \varphi_{d-1}^{\text{ReLU}}(\mathbf{q}_{d-1}; \kappa_{d-1}; n_1, \ldots, n_{d-2}).
\end{aligned}
\tag{B.4}
$$

By the induction hypothesis, we have

$$
\begin{aligned}
\varphi_{d-1}^{\text{ReLU}}&(\mathbf{q}_{d-1}; \kappa_{d-1}; n_1, \ldots, n_{d-2}) \\
&= \frac{2^{n_1 + \cdots + n_{d-2}} - (2^{n_1} - 1)(2^{n_2} - 1) \cdots (2^{n_{d-2}} - 1)}{2^{n_1 + \cdots + n_{d-2}}} \\
&\quad + \frac{1}{2^{n_1 + \cdots + n_{d-2}}} \sum_{k_1=1}^{n_1} \cdots \sum_{k_{d-2}=1}^{n_{d-2}} \binom{n_1}{k_1} \cdots \binom{n_{d-2}}{k_{d-2}} \varphi_{d-1}^{\text{lin}}(\mathbf{q}_{d-1}; \kappa_{d-1}; k_1, \ldots, k_{d-2}).
\end{aligned}
\tag{B.5}
$$

Noting that

$$
\int \frac{d\mathbf{q}_{d-1} \, d\mathbf{h}_{d-1}}{(2\pi)^{n_{d-1}}} \exp\left( i\mathbf{q}_{d-1} \cdot \mathbf{h}_{d-1} - \frac{1}{2} \sigma_d^2 \|\mathbf{q}_d\|^2 \|\phi(\mathbf{h}_{d-1})\|^2 \right) = 1,
\tag{B.6}
$$

our task is to evaluate the integral

$$
\int \frac{d\mathbf{q}_{d-1} \, d\mathbf{h}_{d-1}}{(2\pi)^{n_{d-1}}} \exp\left( i\mathbf{q}_{d-1} \cdot \mathbf{h}_{d-1} - \frac{1}{2} \sigma_d^2 \|\mathbf{q}_d\|^2 \|\phi(\mathbf{h}_{d-1})\|^2 \right) \varphi_{d-1}^{\text{lin}}(\mathbf{q}_{d-1}; \kappa_{d-1}; k_1, \ldots, k_{d-2}).
\tag{B.7}
$$

By definition,

$$
\begin{aligned}
\int \frac{d\mathbf{q}_{d-1}}{(2\pi)^{n_{d-1}}} \exp(i\mathbf{q}_{d-1} \cdot \mathbf{h}_{d-1}) \varphi_{d-1}^{\text{lin}}(\mathbf{q}_{d-1}; \kappa_{d-1}; k_1, \ldots, k_{d-2}) \\
= p_{d-1}^{\text{lin}}(\mathbf{h}_{d-1}; \kappa_{d-1}; k_1, \ldots, k_{d-2}, n_{d-1}),
\end{aligned}
\tag{B.8}
$$

hence the required integral is

$$\int d\mathbf{h}_{d-1} \, \exp\left(-\frac{1}{2}\sigma_d^2 \|\mathbf{q}_d\|^2 \|\phi(\mathbf{h}_{d-1})\|^2\right) p_{d-1}^{\text{lin}}(\mathbf{h}_{d-1}; \kappa_{d-1}; k_1, \ldots, k_{d-2}, n_{d-1}). \quad \text{(B.9)}$$

As $p_{d-1}^{\text{lin}}$ is radial, the integral is invariant under permutation of the dimensions of $\mathbf{h}_{d-1}$. Then, partitioning the domain of integration over $\mathbf{h}_2$ into regions in which different numbers of ReLUs are active, we have

$$\sum_{k_{d-1}=0}^{n_{d-1}} \binom{n_{d-1}}{k_{d-1}} \int_0^\infty dh_{d-1,1} \cdots \int_0^\infty dh_{d-1,k_{d-1}} \, \exp\left(-\frac{1}{2}\sigma_d^2\|\mathbf{q}_d\|^2 \sum_{j=1}^{k_{d-1}} h_{d-1,j}^2\right)$$
$$\times \int_{-\infty}^0 dh_{d-1,k_{d-1}+1} \cdots \int_{-\infty}^0 dh_{d-1,n_{d-1}} \, p_{d-1}^{\text{lin}}(\mathbf{h}_{d-1}; \kappa_{d-1}; k_1, \ldots, k_{d-2}, n_{d-1}). \tag{B.10}$$

As the integrand is even in each dimension of $\mathbf{h}_{d-1}$, we can extend the domain of integration to all of $\mathbb{R}^{n_{d-1}}$ at the expense of a factor of $2^{-n_{d-1}}$:

$$\frac{1}{2^{n_{d-1}}} \sum_{k_{d-1}=0}^{n_{d-1}} \binom{n_{d-1}}{k_{d-1}} \int_{-\infty}^\infty dh_{d-1,1} \cdots \int_{-\infty}^\infty dh_{d-1,k_{d-1}} \, \exp\left(-\frac{1}{2}\sigma_d^2\|\mathbf{q}_d\|^2 \sum_{j=1}^{k_{d-1}} h_{d-1,j}^2\right)$$
$$\times \int_{-\infty}^\infty dh_{d-1,k_{d-1}+1} \cdots \int_{-\infty}^\infty dh_{d-1,n_{d-1}} \, p_{d-1}^{\text{lin}}(\mathbf{h}_{d-1}; \kappa_{d-1}; k_1, \ldots, k_{d-2}, n_{d-1}). \tag{B.11}$$

We now use the fact that

$$\int_{-\infty}^\infty dh_{d-1,k_{d-1}+1} \cdots \int_{-\infty}^\infty dh_{d-1,n_{d-1}} \, p_{d-1}^{\text{lin}}(\mathbf{h}_{d-1}; \kappa_{d-1}; k_1, \ldots, k_{d-2}, n_{d-1})$$
$$= p_{d-1}^{\text{lin}}(\mathbf{h}_{d-1}; \kappa_{d-1}; k_1, \ldots, k_{d-2}, k_{d-1}), \tag{B.12}$$

which, as noted in the main text, follows from its definition. Next, we note that

$$\int_{-\infty}^\infty dh_{d-1,1} \cdots \int_{-\infty}^\infty dh_{d-1,k_{d-1}} \, \exp\left(-\frac{1}{2}\sigma_d^2\|\mathbf{q}_d\|^2 \sum_{j=1}^{k_{d-1}} h_{d-1,j}^2\right)$$
$$\times p_{d-1}^{\text{lin}}(\mathbf{h}_{d-1}; \kappa_{d-1}; k_1, \ldots, k_{d-2}, k_{d-1})$$
$$= \varphi_d^{\text{lin}}(\mathbf{q}_d; \kappa_{d-1}; k_1, \ldots, k_{d-1}) \tag{B.13}$$

by the recursive relationship between the characteristic functions. If $k_{d-1} = 0$, this quantity is replaced by unity. Thus, the integral of interest evaluates to

$$\frac{1}{2^{n_{d-1}}} + \frac{1}{2^{n_{d-1}}} \sum_{k_{d-1}=0}^{n_{d-1}} \binom{n_{d-1}}{k_{d-1}} \varphi_d^{\text{lin}}(\mathbf{q}_d; \kappa_{d-1}; k_1, \ldots, k_{d-1}). \tag{B.14}$$

Therefore, after some algebraic simplification of the constant term, we find that

$$\varphi_d(\mathbf{q}_d; \kappa_d; n_1, \ldots, n_{d-1})$$
$$= 1 - \frac{(2^{n_1} - 1)(2^{n_2} - 1) \cdots (2^{n_{d-1}} - 1)}{2^{n_1 + \cdots + n_{d-1}}}$$
$$+ \frac{1}{2^{n_1 + \cdots + n_{d-1}}} \sum_{k_1=1}^{n_1} \cdots \sum_{k_{d-1}=1}^{n_{d-1}} \binom{n_1}{k_1} \cdots \binom{n_{d-1}}{k_{d-1}} \varphi_d^{\text{lin}}(\mathbf{q}_d; \kappa_{d-1}; k_1, \ldots, k_{d-1}) \quad \text{(B.15)}$$

under the induction hypothesis, hence we conclude the claimed result.

# C   Derivation of tail bounds

In this appendix, we use our results for the moments of the preactivation norms to derive the variation of the tail bounds of [7, 8] reported in §4.2. Following the results of Vladimirova et al. [7, 8], it suffices to show that there exist positive constants $C_1$ and $C_2$ such that

$$C_1 m^{d/2} \leq (\mathbb{E}\|\mathbf{h}_d\|^m)^{1/m} \leq C_2 m^{d/2} \tag{C.1}$$

for all $m \in \mathbb{N}_{>0}$, holding the widths $n_1, \ldots, n_d$ and the depth $d$ fixed. It is of course sufficient to show that $(\mathbb{E}\|\mathbf{h}_d\|^m)^{1/m}$ behaves asymptotically like $m^{d/2}$, as the constants $C_1$ and $C_2$ may be chosen small and large enough, respectively, such that this inequality holds for smaller, finite $m$.

For a linear network, we have (17)

$$(\mathbb{E}_{\mathrm{lin}}\|\mathbf{h}_d\|^m)^{1/m} = 2^{d/2}\kappa_d \prod_{\ell=1}^{d} \left( \frac{\Gamma[(n_\ell + m)/2]}{\Gamma(n_\ell/2)} \right)^{1/m}. \tag{C.2}$$

By a simple application of Stirling's formula [2], we find that

$$\left( \frac{\Gamma[(n + m)/2]}{\Gamma(n/2)} \right)^{1/m} = \sqrt{\frac{m}{2e}}[1 + \mathcal{O}(m^{-1})] \tag{C.3}$$

as $m \to \infty$ for any fixed $n \in \mathbb{N}_{>0}$. Therefore, for any finite depth, we conclude the desired result.

For a ReLU network, we have (18)

$$(\mathbb{E}_{\mathrm{ReLU}}\|\mathbf{h}_d\|^m)^{1/m} = 2^{d/2}\kappa_d \left( \frac{\Gamma[(n_d + m)/2]}{\Gamma(n_d/2)} \right)^{1/m} \prod_{\ell=1}^{d-1} \left[ \frac{1}{2^{n_\ell}} \sum_{k_\ell=1}^{n_\ell} \binom{n_\ell}{k_\ell} \frac{\Gamma[(k_\ell + m)/2]}{\Gamma(k_\ell/2)} \right]^{1/m}. \tag{C.4}$$

Trivially,

$$\frac{1}{2^n} \sum_{k=1}^{n} \binom{n}{k} \frac{\Gamma[(k + m)/2]}{\Gamma(k/2)} \leq (1 - 2^n) \frac{\Gamma[(n + m)/2]}{\Gamma(n/2)} \leq \frac{\Gamma[(n + m)/2]}{\Gamma(n/2)}. \tag{C.5}$$

Similarly, we have the trivial lower bound

$$\frac{1}{2^n} \sum_{k=1}^{n} \binom{n}{k} \frac{\Gamma[(k + m)/2]}{\Gamma(k/2)} \geq (1 - 2^n) \frac{\Gamma[(1 + m)/2]}{\Gamma(1/2)}, \tag{C.6}$$

hence, as $(1 - 2^n)^{1/m} \geq 1/2$ for all $m, n \in \mathbb{N}_{>0}$, we have

$$\frac{1}{2} \left( \frac{\Gamma[(1 + m)/2]}{\Gamma(1/2)} \right)^{1/m} \leq \left( \frac{1}{2^n} \sum_{k=1}^{n} \binom{n}{k} \frac{\Gamma[(k + m)/2]}{\Gamma(k/2)} \right)^{1/m} \leq \left( \frac{\Gamma[(n + m)/2]}{\Gamma(n/2)} \right)^{1/m}. \tag{C.7}$$

Thus, by virtue of the above result for linear networks, we obtain the desired result.

# D   Derivation of the asymptotic prior distribution at large widths

In this appendix, we derive the asymptotic behavior of the prior distribution for large hidden layer widths reported in §4.3. We first consider linear networks. We assume the parameterization described in the main text, which yields

$$\mathbb{E}h_i h_j = \varkappa_d^2 \delta_{ij} \tag{D.1}$$

for $\varkappa_d$ independent of width. Then, using the fact that all odd-ordered cumulants of the zero-mean random vector $\mathbf{h}_d$ vanish, the third-order Edgeworth approximation to the prior is

$$p_d(\mathbf{h}_d \,|\, \mathbf{x}) \approx \frac{1}{(2\pi\varkappa_d^2)^{n_d/2}} \exp\left( -\frac{\|\mathbf{h}_d\|^2}{2\varkappa_d^2} \right)$$
$$\times \left[ 1 + \frac{1}{24}\chi_{ijkl} \left( \frac{1}{\varkappa_d^8} h_i h_j h_k h_l - \frac{6}{\varkappa_d^6} \delta_{kl} h_i h_j + \frac{3}{\varkappa_d^2} \delta_{ij}\delta_{kl} \right) \right], \tag{D.2}$$

where

$$\chi_{ijkl} = \mathbb{E}h_i h_j h_k h_l - \mathbb{E}(h_i h_j)\mathbb{E}(h_k h_l) - \mathbb{E}(h_i h_k)\mathbb{E}(h_j h_l) - \mathbb{E}(h_i h_l)\mathbb{E}(h_j h_k) \tag{D.3}$$

is the fourth joint cumulant and summation over repeated indices is implied [9]. For this Edgeworth approximation to yield an asymptotic approximation to the prior (i.e., for higher terms to be suppressed in the limit of large widths), the sixth and higher cumulants of $\mathbf{h}_d$ must be suppressed relative to the fourth cumulant. However, using the radial symmetry of the distribution and the moments (17), we can see that these cumulants will be of $\mathcal{O}(n^{-2})$.

We now note that the only non-vanishing terms will be those of the form $\chi_{iiii}$, $\chi_{iijj}$, $\chi_{ijij}$, or $\chi_{iijj}$, and that

$$\chi_{iiii} = \mathbb{E}h_i^4 - 3(\mathbb{E}h_i^2)^2, \tag{D.4}$$

while

$$\chi_{iijj} = \chi_{ijij} = \chi_{iijj} = \mathbb{E}h_i^2 h_j^2 - \mathbb{E}(h_i^2)\mathbb{E}(h_j^2). \tag{D.5}$$

By symmetry or by direct calculation in spherical coordinates, we have

$$\mathbb{E}h_1^4 = 3\mathbb{E}h_1^2 h_2^2 = 3\kappa_d^4 \prod_{\ell=1}^{d-1}[n_\ell(n_\ell+2)] = 3\varkappa_d^4 \prod_{\ell=1}^{d-1}\frac{n_\ell+2}{n_\ell}, \tag{D.6}$$

hence

$$\chi_{iiii} = 3\chi_{iijj} = 3\varkappa_d^4 \left[\prod_{\ell=1}^{d-1}\frac{n_\ell+2}{n_\ell} - 1\right]. \tag{D.7}$$

Therefore, approximating $\chi_{iiii}$ to $\mathcal{O}(n^{-1})$, we obtain the following third-order Edgeworth approximation for the prior density:

$$
\begin{aligned}
p_d(\mathbf{h}_d \,|\, \mathbf{x}) \approx{}& \frac{1}{(2\pi\varkappa_d^2)^{n_d/2}} \exp\left(-\frac{\|\mathbf{h}_d\|^2}{2\varkappa_d^2}\right) \\
&\times \left[1 + \frac{1}{4}\left(\sum_{\ell=1}^{d-1}\frac{1}{n_\ell}\right)\left(\frac{\|\mathbf{h}_d\|^4}{\varkappa_d^4} - 2(n_d+2)\frac{\|\mathbf{h}_d\|^2}{\varkappa_d^2} + n_d(n_d+2)\right) + \mathcal{O}\left(\frac{1}{n^2}\right)\right].
\end{aligned}
\tag{D.8}
$$

Upon integration, the second term inside the square brackets vanishes, hence this approximate density is properly normalized.

For ReLU networks, the story is much the same, except we now have $\mathbb{E}h_i h_j = 2^{1-d}\varkappa_d^2 \delta_{ij}$ and

$$\mathbb{E}h_1^4 = 3\mathbb{E}h_1^2 h_2^2 = 3 \times 4^{1-d}\kappa_d^4 \prod_{\ell=1}^{d-1}[n_\ell(n_\ell+5)] = 3 \times 4^{1-d}\varkappa_d^4 \prod_{\ell=1}^{d-1}\frac{n_\ell+5}{n_\ell}, \tag{D.9}$$

hence we conclude that

$$
\begin{aligned}
p_d^{\text{ReLU}}(\mathbf{h}_d \,|\, \mathbf{x}) \approx{}& \frac{1}{(2^{2-d}\pi\varkappa_d^2)^{n_d/2}} \exp\left(-\frac{\|\mathbf{h}_d\|^2}{2^{2-d}\varkappa_d^2}\right) \\
&\times \left[1 + \frac{5}{4}\left(\sum_{\ell=1}^{d-1}\frac{1}{n_\ell}\right)\left(\frac{\|\mathbf{h}_d\|^4}{4^{1-d}\varkappa_d^4} - 2(n_d+2)\frac{\|\mathbf{h}_d\|^2}{2^{1-d}\varkappa_d^2} + n_d(n_d+2)\right)\right. \\
&\left. \quad + \mathcal{O}\left(\frac{1}{n^2}\right)\right].
\end{aligned}
\tag{D.10}
$$

One can immediately see that these approximate distributions are sub-Gaussian. To show this more formally, we note that the moments of the approximate distribution for a linear network are

$$(\mathbb{E}_{\text{EW}}\|\mathbf{h}_d\|^m)^{1/m} = \sqrt{2}\varkappa_d \left(\frac{\Gamma[(n_d+m)/2]}{\Gamma(n_d/2)}\right)^{1/m}\left[1 + \frac{1}{4}\left(\prod_{\ell=1}^{d-1}\frac{1}{n_\ell}\right)m(m-2)\right]^{1/m}. \tag{D.11}$$

For all $m \geq 2$ and $0 \leq t \leq 1$, we have

$$1 \leq [1 + m(m-2)t]^{1/m} \leq (m-1)^{2/m} \leq 2, \tag{D.12}$$

where the upper bound is sub-optimal but sufficient for our purposes. Then, we conclude that

$$\sqrt{2}\varkappa_d \left( \frac{\Gamma[(n_d + m)/2]}{\Gamma(n_d/2)} \right)^{1/m} \leq (\mathbb{E}_{\mathrm{EW}} \|\mathbf{h}_d\|^m)^{1/m} \leq 2\sqrt{2}\varkappa_d \left( \frac{\Gamma[(n_d + m)/2]}{\Gamma(n_d/2)} \right)^{1/m} \tag{D.13}$$

for all $m \geq 2$. Moreover, we can easily see that similar bounds will hold for the approximation to the prior of a ReLU network, up to overal factors scaling $\varkappa_d$. Therefore, applying the results of Appendix C, we conclude that these approximations are sub-Weibull with optimal tail exponent $1/2$, implying that they are sub-Gaussian.

## E   Numerical methods

Here, we summarize the numerical methods used to generate Figures 1-4. All computations were performed using MATLAB versions 9.5 (R2018b) and 9.8 (R2020a).[1] The theoretical prior densities were computed using the `meijerG` function, and evaluated with variable-precision arithmetic. Empirical distributions were estimated with simple Monte Carlo sampling: for each sample, the weight matrices were drawn from isotropic Gaussian distributions, and then the output preactivation was computed. In these simulations, the input was taken to be one-dimensional and to have a value of unity. Furthermore, we fixed $\kappa_d^2 = (n_1 \cdots n_{d-1})^{-1}$ for linear networks and $\kappa_d^2 = 2^{d-1}(n_1 \cdots n_{d-1})^{-1}$ for ReLU networks, such that the output preactivations had identical variances.

The computations required to evaluate the theoretical priors and sampling-based estimates in Figures 1 and 3 were performed across 32 CPU cores of one node of Harvard University's Cannon HPC cluster.[2] The computational cost of our work was entirely dominated by evaluation of the theoretical ReLU network prior. To reduce the amount of computation required to evaluate the ReLU network prior at large widths, we approximated the full mixture (16) by neglecting terms with weighting coefficients $2^{-n_\ell}\binom{n_\ell}{k_\ell}$ less than the floating-point relative accuracy `eps` $= 2^{-52}$. More precisely, our code evaluates the logarithm of the weighting coefficient using the $\log \Gamma$ function (`gammaln` in MATLAB) for numerical stability, and then compares the logarithms of these two non-negative floating point values. This cutoff only truncates the sum for networks of width $n = 100$ at depths $d = 2, 3,$ and $4$; the full mixture is evaluated for narrower networks. For $n = 100$, it reduces the number of summands from $10^2$, $10^4$, and $10^6$ to 77, 4,537, and 208,243, respectively. We have confirmed that the resulting approximation to the exact prior behaves monotonically with respect to the cutoff for values larger than `eps`. With this cutoff, 24 seconds, 3.5 hours, and 153 hours of compute time were required to compute the theoretical prior for these depths, respectively. In all, we required just under 160 hours of compute time to produce the figures shown here.