# OpenReview forum: "Exact marginal prior distributions of finite Bayesian neural networks"
_NeurIPS.cc/2021/Conference — NeurIPS 2021 Spotlight_

### Official Review · Reviewer_yWcY · 2021-07-16

**Rating:** 7
**Confidence:** 3

**Summary:**

This paper derives closed from expressions for the marginal distribution of full-connected ReLU neural networks, with Gaussian distributions over weights, and no biases. (Can be viewed as Bayesian NN priors or NNs at initialisation.) Importantly, this is done for finite widths. This contrasts with much previous work which assumes infinite width (-> GP). It unifies/confirms some results on finite width networks.

**Limitations And Societal Impact:**

Yes. Was done well.

**Main Review:**

The paper offers a worthwhile contribution, is well written, and seems technically sound (disclaimer: I have not reviewed the mathematical derivations in appendix A & B). I can see it becoming well cited and useful for future work.

The main limitation is the reduced scope considered by the paper – fully-connected networks, ReLU non-linearities, no biases, Gaussian weight distributions. Also, it’s only the marginal distribution that’s considered – not the covariance between two inputs as is often of interest in the BNN/GP literature. That said it’s a useful theoretical result in itself.

Whilst the paper was good at reminding in several places this limited scope, the current title risks being misleadingly general. I’d like to suggest aligning the paper’s title with the scope, something like: “Exact conditional distributions of finite neural networks at initialisation” or “Exact marginal prior distributions of finite Bayesian neural networks”, which I think is a more honest representation of the contents of the paper. I’d be interested to get other reviewers’ opinions on this.

Whilst this is a theoretical paper, it would be nice to discuss possible potential impacts of the results. Is the heavy-tailedness of deeper networks a problem? I appreciated the discussion on the cold posterior, but wasn’t clear exactly how the paper’s results contribute to that issue? Are there other consequences, e.g. ideas for initialisation schemes?


__Minor__

Perhaps denote eq 14 as p^lin_d since this is used later on?

**Time Spent Reviewing:**

4

---

> ### Author Response · Authors · 2021-08-09
> **Response to Reviewer yWcY**
>
> *[...]*
>
> *Whilst the paper was good at reminding in several places this limited scope, the current title risks being misleadingly general. I’d like to suggest aligning the paper’s title with the scope, something like: “Exact conditional distributions of finite neural networks at initialisation” or “Exact marginal prior distributions of finite Bayesian neural networks”, which I think is a more honest representation of the contents of the paper. I’d be interested to get other reviewers’ opinions on this.*
>
> Thank you for this comment. As detailed in our common response, we intend to adopt your suggestion of “Exact marginal prior distributions of finite Bayesian neural networks.”
>
> *Whilst this is a theoretical paper, it would be nice to discuss possible potential impacts of the results. Is the heavy-tailedness of deeper networks a problem? I appreciated the discussion on the cold posterior, but wasn’t clear exactly how the paper’s results contribute to that issue? Are there other consequences, e.g. ideas for initialisation schemes?*
>
> Thank you for this suggestion. As described in our discussion of significance under "Common concerns," we will provide a more detailed discussion of possible impacts of our results in our revised manuscript. In particular, we will clarify how our results might contribute to understanding the issue of cold posteriors.
>
> ***Minor***
>
> *Perhaps denote eq 14 as $p^{lin}_d$ since this is used later on?*
>
> Thank you for this suggestion; we will use this notation for the linear network prior throughout.

---

> > ### Comment · Reviewer_yWcY · 2021-08-19
> > **Post-rebuttal response**
> >
> > Thanks for the response.

---

### Official Review · Reviewer_ZUme · 2021-07-16

**Rating:** 6
**Confidence:** 3

**Summary:**

While infinitely wide Bayesian neural networks (BNNs) hidden units distributions are studied well enough due to the Gaussian process limit, hidden units at finite widths are now under the telescope. The main result of the paper is a derivation of the exact characterization of priors in function space through Meijer G-functions. The results are obtained for Gaussian priors, ReLU and linear activation functions.

**Advantages**:
* It is the first accurate description of hidden units distributions.
* The results are in line with other works on the heavy-tailed nature of hidden units, non-Gaussian corrections, and bounded moments.

**Limitations**:
* Only linear and ReLU functions,
* Only Gaussian priors,
* Without the bias term.

**Limitations And Societal Impact:**

The authors have adequately addressed the limitations and potential negative societal impact of their work

**Main Review:**

**Originality**:
The paper is original and the first to find an accurate description of hidden units distributions. There is also a concurrent work called “Precise characterization of the prior predictive distribution of deep ReLU networks” that appeared after the submission and I guess the authors have seen it already.

**Clarity**:

I find the paper well-written and clear.

**Conclusion**:

Though the results are obtained for a simple model and it seems hard to generalize to other settings or priors, I find the work worth publishing.


**Minor**:

* The notation “output prior” seems not accurate for me. As it is about hidden units outputs, I would precise or change to one of following: “function space prior”, “prior predictive”, “hidden units prior”.
* “Our approach complements to the study” -> without to

**Time Spent Reviewing:**

16

---

> ### Author Response · Authors · 2021-08-09
> **Response to Reviewer ZUme**
>
> *[...]*
>
> ***Originality:*** *The paper is original and the first to find an accurate description of hidden units distributions. There is also a concurrent work called “Precise characterization of the prior predictive distribution of deep ReLU networks” that appeared after the submission and I guess the authors have seen it already.*
>
> Thank you for this comment. We saw the "Precise characterization..." paper when it appeared on the arXiv after the submission deadline, and will acknowledge it in the revised version of our manuscript.
>
> *[...]*
>
> ***Minor:***
>
> *The notation “output prior” seems not accurate for me. As it is about hidden units outputs, I would precise or change to one of following: “function space prior”, “prior predictive”, “hidden units prior”.*
>
> Thank you for this suggestion. We had initially avoided using the terminology "function space prior" due to the fact that, in physics, this phrasing would suggest the measure appearing in a functional integral rather than the measure in an integral over function evaluations on a fixed input. However, as reviewer **jHrx** also uses this terminology, we would be happy to adopt it in the revised paper.
>
> *"Our approach complements to the study" -> without to*
>
> Thank you for catching this typo; we've fixed it.

---

### Official Review · Reviewer_jHrx · 2021-07-16

**Rating:** 8
**Confidence:** 4

**Summary:**

The paper calculates the function-space prior (marginally for single outputs) induced by an independent Gaussian prior on the weights for deep linear and ReLU finite NNs without biases. This prior is expressed using the ​​Meijer G-function for deep linear networks, and weighted sums of these functions for deep ReLU networks. This makes clear that at finite widths, the prior is increasingly heavy-tailed with increasing depths. Moreover, finite-width corrections are unable to capture this heavy-tailedness as it is not a perturbative effect.

**Limitations And Societal Impact:**

Yes.

**Main Review:**


Originality: This is the first exact characterization of the function-space prior of finite-width NN that I know of. While the expression of that prior is complicated, relating to a preexisting class of special functions is useful for theorists. “Deep Neural Networks as Gaussian Processes” Lee et al. should be cited in the introduction.

Quality: The high level argument presented in the main text is reasonable, but I did not read the proof in the supplement in detail. The results seem correct and are supported by experimental evidence in the figures.
Although it might be out of scope for this paper, the statistical implications of the heavy-tailed of the prior are under explored. Some experiments showing how this leads to improved performance on heavy-tailed data would be interesting. Similarly, are there any benefits to inference from knowing the exact form of the prior for finite widths (other than the improved understanding of the prior induced by the hyperparameters)?

Clarity: The calculation of the priors is technical but the major points are presented clearly in the main text. The figures are clear and emphasize the main points of the paper, although they are quite similar. The paper might be improved if the authors presented this information in alternate ways for different figures.

- Eq. (6) has a typo: h_2 -> h_{d-2}
- What is the dash in Eq. (14)?

Significance: This result is important and will be of interest to theorists at NeurIPS. There are many interesting follow up questions.

**Time Spent Reviewing:**

6

---

> ### Author Response · Authors · 2021-08-09
> **Response to Reviewer jHrx**
>
> ***Originality:*** *This is the first exact characterization of the function-space prior of finite-width NN that I know of. While the expression of that prior is complicated, relating to a preexisting class of special functions is useful for theorists. “Deep Neural Networks as Gaussian Processes” Lee et al. should be cited in the introduction.*
>
> Thank you for this comment. As mentioned in our response to Reviewer **XUQc**, we will cite Lee et al. in the revised version of our manuscript.
>
> ***Quality:*** *[...] Although it might be out of scope for this paper, the statistical implications of the heavy-tailed of the prior are under explored. Some experiments showing how this leads to improved performance on heavy-tailed data would be interesting. Similarly, are there any benefits to inference from knowing the exact form of the prior for finite widths (other than the improved understanding of the prior induced by the hyperparameters)?*
>
> Thank you for these suggestions. As described in our detailed discussion of "Significance and implications" under "Common concerns," knowing the exact form of the prior allows one to gain an intuitive understanding of the posterior for a given likelihood. In particular, we agree that studying how heavy-tailed priors might lead to improved performance on heavy-tailed data (relative to thin-tailed priors, of course) is an interesting objective, though we will reserve detailed experimental studies for future work.
>
> ***Clarity:*** *[...] The figures are clear and emphasize the main points of the paper, although they are quite similar. The paper might be improved if the authors presented this information in alternate ways for different figures.*
>
> Thank you for this suggestion. To help differentiate Figures 2 and 3, we will add a diagram to Figure 3 that illustrates the mixture-of-linear-networks interpretation of (16).
>
> *Eq. (6) has a typo: $h_2$ -> $h_{d-2}$*
>
> Thank you, fixed.
>
> *What is the dash in Eq. (14)?*
>
> The dash in (14) indicates the absence of any "upper" arguments in $G^{d,0}_{0,d}$, i.e., $a_{1},\ldots,a_{p}$ in eq. (12). We will add a comment clarifying the meaning of this notation.

---

> > ### Comment · Reviewer_jHrx · 2021-08-16
> > **Follow up**
> >
> > While there are limitations and clear follow up questions point out by the reviewers, I don't think there are any substantial issues with the paper. I'm happy with the authors' answers to be concerns and questions, and will raise my score accordingly.

---

### Official Review · Reviewer_XUQc · 2021-08-01

**Rating:** 6
**Confidence:** 5

**Summary:**

This paper considers random finite-width, finite-depth fully-connected neural networks and derives exact formulas for the probability distribution for neural network preactivations, conditioned on a single input, at initialization. The cases considered include deep linear networks and deep Relu networks without biases. These prior distributions are expressable in terms of Meijer G-functions and are rotationally invariant (dependent only on the magnitude of preactivations). The exact priors are visualized for different widths and depths and compared to exact results for infinite-width networks (where the result is a Gaussian distribution) and perturbative results for large but finite-width networks (where the result can be expressed using an Edgeworth series). Notably, the exact results -- which are non-perturbative and have no constraints on depth and width -- reveal a long and heavy tail in the distribution which is not evident in the large width limit.

**Limitations And Societal Impact:**

The authors are transparent about the limitations. Omitting discussion of societal impact is ok for this type of paper.

**Main Review:**

Originality: The paper addresses a timely research topic -- trying to theoretically characterize the prior and posterior distributions of deep neural networks. Previous work has largely used other methodology; the technical derivations in this paper (e.g. connection to Meijer G-functions) seem original, to the best of my knowledge.

Quality: The quality of the work appears solid, and I could not identify any mistakes (however, I have not checked the derivations in detail, specifically the appendix).

Clarity: The paper is well written & paced and straightforward to read.

Significance: The relative significance of this paper is my reason for a slightly lower score. I think the paper makes a nice and publishable (in some form) contribution to the literature on this topic, but I am concerned it may be of limited interest relative to other papers publishable at NeurIPS. The contexts in which the authors are able to derive results is a bit narrow in scope: a single input (multiple input case is left for future work), no biases (this is perhaps not of greatest concern, however), and treating only the prior distribution. It is illuminating to see that the exact results display heavy tails and are qualitatively different from large width results, although similar insight has appear in previous work (Vladimirova, et al).

Other comments:
--Given that the scope of the work is limited to a single input, I would suggest a small modification to the paper title that somehow makes this clear
--Another work that is of some relevance (since they consider the case of a single input and marginalization over intermediate variables) is https://arxiv.org/abs/1710.06570.
--A paper with overlapping results to Matthews, et al. (Ref 5) is Lee, et al. https://arxiv.org/abs/1711.00165 and is missing from the citations.
--Could the authors comment on the connection of their work to the literature on deep Gaussian processes?
--Is the notation on L113 intended as written? ({a} and {b} series ranging from 1 ... p and no 1 ... q)
--In addition to Ref 34 & 35 cited in L228, I believe Ref 10 also considered corrections to the finite-width posterior / inference.

**Time Spent Reviewing:**

5

---

> ### Author Response · Authors · 2021-08-09
> **Response to Reviewer XUQc**
>
> *[...]*
>
> ***Other comments:***
>
> - *Given that the scope of the work is limited to a single input, I would suggest a small modification to the paper title that somehow makes this clear*
>
>     As we describe in detail in our common response, we propose to change the paper title to "Exact marginal prior distributions of finite Bayesian neural networks." We appreciate your feedback on whether this makes the scope of our work sufficiently clear.
>
> - *Another work that is of some relevance (since they consider the case of a single input and marginalization over intermediate variables) is https://arxiv.org/abs/1710.06570.*
>
>     Thank you for this pointer; we will discuss this interesting and relevant paper in our revised manuscript.
>
> - *A paper with overlapping results to Matthews, et al. (Ref 5) is Lee, et al. https://arxiv.org/abs/1711.00165 and is missing from the citations.*
>
>     Thank you for this comment. Our failure to cite Lee et al. was an unintentional oversight, which we will rectify in the revised version of our manuscript.
>
> - *Could the authors comment on the connection of their work to the literature on deep Gaussian processes?*
>
>     Thank you for this suggestion. We will add a sentence to the Introduction of our paper to introduce the fact that finite BNNs are a class of deep GPs, and comment on the study of deep GP priors by Lu et al. (https://arxiv.org/abs/1905.10963), who showed using moment computations that the function-space priors become increasingly heavy-tailed with depth. Moreover, we will add a discussion of closely related work on deep GP limits of networks with finite bottlenecks separating infinitely-wide layers by Agrawal et al. (https://jmlr.org/papers/v21/20-017.html). We welcome further feedback from the referee on this point.
>
> - *Is the notation on L113 intended as written? ({a} and {b} series ranging from 1 ... p and no 1 ... q)*
>
>     Thank you for catching this typo; the indices for $\{b_{k}\}$ should range from $1$ to $q$ (not $1$ to $p$), as indicated by (12).
>
> - *In addition to Ref 34 & 35 cited in L228, I believe Ref 10 also considered corrections to the finite-width posterior / inference.*
>
>     We now cite Ref. 10 along with 34 and 35 in this sentence.

---

> > ### Comment · Reviewer_XUQc · 2021-08-23
> > **follow-up**
> >
> > Thank you to the authors for their reply.
> >
> > --The proposed title change "Exact marginal prior distributions of finite Bayesian neural networks" sounds good to me.
> >
> > --Thank you for including the results from the deep GP literature in the revision; I have no further suggestions here.

---

### Author Response · Authors · 2021-08-09
**Common concerns for "Exact priors of finite neural networks"**

We thank all referees for their thoughtful reviews of our paper. In this comment, we reply to common concerns; specific comments are addressed individually below.

***Paper title***: Reviewers **XUQc** and **yWcY** suggest that the paper title be modified to more precisely reflect its scope. As we certainly wish to avoid the possibility that readers might misapprehend the generality of our results, we will do so. We are inclined to accept Reviewer **yWCY**'s suggestion of "Exact marginal prior distributions of finite Bayesian neural networks." We welcome feedback from other referees on whether this proposed alternative would adequately address their concerns.

***Significance and implications***: Reviewers **XUQc**, **jHrx**, and **yWcY** all indicated that they would appreciate more extensive discussion of the significance and implications of our results, given that they are derived in a fairly restrictive setting. As noted by the referees, the primary contribution of our results is a finer-grained understanding of how hyperparameter choice affects the function space prior than that afforded by previous work (such as Vladimirova et al.). In our revised manuscript, we will revise our discussion to clarify how knowledge of the exact form of the prior might contribute to understanding inference and the cold posterior effect. In particular, though we do not compute function-space posterior distributions, knowing the precise form of the prior would allow one to gain an intuitive understanding of the shape of the posterior for a given likelihood (even without computing the normalization factor resulting from integrating the product of the prior and likelihood over the function values). For instance, as suggested by Reviwer **jHrx**, one could imagine a particular degree of heavy-tailedness in the prior being optimal for a dataset that is to some degree heavy-tailed. This could allow one to gain some intuition for when the prior or likelihood is misspecified for a given dataset; such misspecification underlies cold posterior effects. Detailed experimental and analytical investigation of these questions is an important objective of our future work. Moreover, we would like to emphasize that the restriction to the marginal prior for a single input is not specific to our methods. Even in perturbative studies (such as the original work by Yaida [10] and his subsequent monograph with Roberts, https://arxiv.org/abs/2106.10165) where one can obtain formal expressions for multiple inputs, one must often specialize to a single input in order to make analytical progress beyond one hidden layer.

---

### Decision · Program_Chairs · 2021-09-27

**Decision:**

Accept (Spotlight)

**Comment:**

This paper derives an exact solution for function-space prior expression (marginally for single outputs) induced by an independent Gaussian prior on the weights for deep linear and ReLU finite fully connected feedforward NNs without biases. This can be viewed as Bayesian NN priors or NNs at initialisation for finite width networks. This prior is expressed using the ​​Meijer G-function for deep linear networks, and weighted sums of these functions for deep ReLU networks. This makes clear that at finite widths, the prior is increasingly heavy-tailed with increasing depths. Moreover, finite-width corrections are unable to capture this heavy-tailedness as it is not a perturbative effect.

All reviewers point out that the paper is well-written and clear. Also work presented in the paper is solid and no noticeable issues or mistakes were identified. Reviewers mention that `this is the first exact characterization of the function-space prior of finite-width NN` (`jHrx`). It’s been noted that the technical methodology advanced using the connection to Meijer G-functions is novel in the deep learning theory community.

There are few weaknesses pointed out. One significant one was the limitation to the single input rather than multi inputs. In this regard, reviewers `XUQc` and `yWcY` raised concern with the current title being too broad and authors agreed to modify it to more precisely describe  the contribution. The authors agreed to modify it as "Exact marginal prior distributions of finite Bayesian neural networks".  Another limitation is requirement for particular architecture (deep linear, ReLu) and weight distribution (`ZUme`, `yWcY`, `XUQc`).

Overall, the AC believes this paper is a timely,  worthwhile and significant step towards characterizing prior distribution of finite networks despite some restrictive settings. AC’s opinion is that the limitations are outweighed by the interesting theoretical outcomes (voiced also by reviewer `yWcY` and `jHrx`)  which opens up new effects (e.g. heavy-tailedness) uncapturable by perturbative corrections. There are lots of interesting questions and follow up work that this paper would ignite, thus it would be worth sharing with the broad NeurIPS audience.